# Distributionally Robust Local Non-parametric Conditional Estimation

**Viet Anh Nguyen**    **Fan Zhang**    **José Blanchet**
Stanford University, United States
{viet-anh.nguyen, fzh, jose.blanchet}@stanford.edu

**Erick Delage**
HEC Montréal, Canada
erick.delage@hec.ca

**Yinyu Ye**
Stanford University, United States
yinyu-ye@stanford.edu

## Abstract

Conditional estimation given specific covariate values (i.e., local conditional estimation or functional estimation) is ubiquitously useful with applications in engineering, social and natural sciences. Existing data-driven non-parametric estimators mostly focus on structured homogeneous data (e.g., weakly independent and stationary data), thus they are sensitive to adversarial noise and may perform poorly under a low sample size. To alleviate these issues, we propose a new distributionally robust estimator that generates non-parametric local estimates by minimizing the worst-case conditional expected loss over all adversarial distributions in a Wasserstein ambiguity set. We show that despite being generally intractable, the local estimator can be efficiently found via convex optimization under broadly applicable settings, and it is robust to the corruption and heterogeneity of the data. Experiments with synthetic and MNIST datasets show the competitive performance of this new class of estimators.

## 1   Introduction

We consider the estimation of conditional statistics of a response variable, $Y \in \mathbb{R}^m$, given the value of a predictor or covariate $X \in \mathbb{R}^n$. The single most important instance of these types of problems involves estimating the conditional mean, or also known as the regression function. Under finite variance assumptions, the conditional mean $\mathbb{E}_{\mathbb{P}}[Y|X = x_0]$ is technically defined as $\psi^{\star}(x_0)$ for some measurable function $\psi^{\star}$ that solves the minimum mean square error problem

$$\min_{\psi} \ \mathbb{E}_{\mathbb{P}}[\|Y - \psi(X)\|_2^2],$$

where the minimization is taken over the space of all measurable functions from $\mathbb{R}^n$ to $\mathbb{R}^m$. While the optimal solution $\psi^{\star}$ is unique up to sets of $\mathbb{P}$-measure zero, unfortunately, solving for $\psi^{\star}$ is challenging because it is an infinite-dimensional optimization problem. The regression function $\psi^{\star}$ can be efficiently found only under specific settings, for example, if one assumes that $(X, Y)$ follows a jointly Gaussian distribution. However, these specific situations are overly restrictive in practice.

In order to bypass the infinite-dimensional challenge involved in directly computing $\psi^{\star}$, we may instead consider a family of optimization problems that are parametrized by $x_0$. More specifically,

in the presence of a regular conditional distribution, the conditional mean $\mathbb{E}_{\mathbb{P}}[Y|X=x_0]$ can be estimated pointwise by $\widehat{\beta}$ defined as

$$\widehat{\beta} \in \arg\min_{\beta} \mathbb{E}_{\mathbb{P}}[\|Y-\beta\|_2^2|X=x_0]$$

for any covariate value $x_0$ of interest. This presents the challenge of effectively accessing the conditional distribution, which is particularly difficult if the event $X=x_0$ has $\mathbb{P}$-probability zero.

Using an analogous argument, if we are interested in the conditional $(\tau \times 100\%)$-quantile of $Y$ given $X$, then this conditional statistics can be estimated pointwise at any location $x_0$ of interest by

$$\widehat{\beta} \in \arg\min_{\beta} \mathbb{E}_{\mathbb{P}}[\max\{-\tau(Y-\beta),(1-\tau)(Y-\beta)\}|X=x_0].$$

The previous examples illustrate that the estimation of a wide range of conditional statistics can be recast into solving a family of finite-dimensional optimization problems parametrically in $x_0$

$$\min_{\beta} \ \mathbb{E}_{\mathbb{P}}[\ell(Y,\beta)|X=x_0] \tag{1}$$

with an appropriately chosen statistical loss function $\ell$.

Problem (1) poses several challenges, some of which were alluded to earlier. First, it requires the integration with respect to a difficult to compute conditional probability distribution. Second, the probability measure $\mathbb{P}$ is generally unknown, hence we lack a fundamental input to solve (1). Finally, in a data-driven setting, there may be few, or even no, observations with value covariate $X=x_0$.

To alleviate these difficulties, our formulation, as we shall explain, involves two features. First, we consider a relaxation of problem (1) in which the event $X=x_0$ is replaced by a neighborhood $\mathcal{N}_{\gamma}(x_0)$ of a suitable radius $\gamma \geq 0$ around $x_0$. Second, we introduce a data-driven distributionally robust optimization (DRO) formulation (e.g. [8, 5, 21]) in order to mitigate the problem that $\mathbb{P}$ is unknown. In turn, the DRO formulation involves a novel class of conditional ambiguity set which copes with the underlying *conditional distribution* being unknown.

In particular, we propose the following *distributionally robust local conditional estimation problem*

$$\min_{\beta} \ \sup_{\mathbb{Q} \in \mathbb{B}_{\rho}^{\infty},\mathbb{Q}(X \in \mathcal{N}_{\gamma}(x_0))>0} \mathbb{E}_{\mathbb{Q}}[\ell(Y,\beta)|X \in \mathcal{N}_{\gamma}(x_0)], \tag{2}$$

where the maximization is taken over all probability measures $\mathbb{Q}$ that are within $\rho$ distance in the $\infty$-Wasserstein sense of a benchmark nominal model, which often corresponds to the empirical distribution of available data. The probability measures $\mathbb{Q}$ are constrained so that $\mathbb{Q}(X \in \mathcal{N}_{\gamma}(x_0)) > 0$ to eliminate the complication of conditioning on a set of measure zero.

**Contributions.** Resting on formulation (2), our main contributions are summarized as follows.

1. We introduce a novel paradigm of non-parametric local *conditional* estimation based on distributionally robust optimization. In contrast to classical non-parametric conditional estimators, our new class of estimators are endowed by design with robustness features. They are structurally built to mitigate the impact of model contamination and therefore they may be reasonably applied to heterogeneous data (e.g., non i.i.d. input).

2. We demonstrate that when the ambiguity set is a type-$\infty$ Wasserstein ball around the empirical measure, the proposed min-max estimation problem can be efficiently solved in many applicable settings, including notably the local conditional mean and quantile estimation.

3. We show that this class of type-$\infty$ Wasserstein local conditional estimators can be considered as a systematic robustification of the $k$-nearest neighbor estimator. We also provide further insights on the statistical properties of our approach and empirical evidence, with both a synthetic and real data sets, that our approach can provide more accurate estimations in practically relevant settings.

**Related work.** One can argue that every single prediction task in machine learning ultimately relates to conditional estimation. So, attempting to provide a full literature survey on non-parametric conditional estimation is an impossible task. Since our contribution is primarily on introducing a novel conceptual paradigm powered by DRO, we focus on discussing well-understood estimators that encompass most of the conceptual ideas used to mitigate the challenges exposed earlier.

The challenges of conditioning on zero probability events and the fact that $x_0$ may not be a part of the sample are addressed based on the idea of averaging around a neighborhood of the point of interest and smoothing. This gives rise to estimators such as $k$-NN (see, for example, [9]), and kernel density estimators, including, for instance the Nadaraya-Watson estimator ([28, 38]) and the Epanechnikov estimator [10], among others. Additional averaging methods include, for example, random forests [6] and Classification and Regression Trees (CARTs, [7]), see also [16] for other techniques.

These averaging and smoothing ideas are well understood, leading to the optimal selection (in a suitable sense) of the kernel along with the associated tuning parameters such as the bandwidth size. These choices are then used to deal with the ignorance of the true data generating distribution by assuming a certain degree of homogeneity in the data, such as stationarity and weak dependence, in order to guarantee consistency and recovery of the underlying generating model. However, none of these estimators are directly designed to cope with the problem of general (potentially adversarial) data contamination.

The later issue revolving around the evaluation of an unknown conditional probability model is connected with robustness, another classical topic in statistics [17]. Much of the classical literature on robustness focuses on the impact of outliers. The work of [41] studies robust-against-outliers kernel regression which enjoys asymptotic consistency and normality under i.i.d. assumptions in a setting where the data contamination becomes negligible. In contrast to this type of contamination, our estimators are designed to be min-max optimal in the DRO sense by supplying the best response against a large (non-parametric) class of adversarial contamination.

Our results can also be seen as connected to adversarial training, which has received a significant amount of attention in recent years [15, 22, 25, 35, 33, 31]. Existing robustification of the nearest neighbors and of the nonparametric classifiers in general can be streamlined into two main strategies: i) global approaches that modify the whole training dataset, e.g., adversarial pruning [37, 40, 4], and ii) local approaches that study well-crafted attack and seek appropriate defense for specific classifiers such as 1-NN [20, 36, 24]. Following this line of ideas, one can interpret our approach as a novel method to train conditional estimators against adversarial attacks. The difference, in the $k$-NN estimation setting for example, is that our attacks are optimal in a distributional sense. Our proposed estimator is thus provably the best for a uniform class of distributional attacks. Compared to the current literature, we believe that our approach is also more general in two significant ways: first, we start from a generic min-max estimation problem, and our ideas and methodology are easily applicable to other non-parametric settings, and second, we allow for perturbations on $Y$ to hedge against label contamination.

DRO-based estimators have generated a great deal of interest because they possess various desirable properties in connection to various forms of regularization (e.g., variance [29]; norm [32]; shrinkage [30]). The tools that we employ are related to those currently being investigated. Our formulation considers adversarial perturbations based on the Wasserstein distance [26, 5, 13, 21]. In particular, the type-$\infty$ Wasserstein distance [14] is recently applied in DRO formulations [1, 3, 39]. In particular, the work of [2] considers adversarial conditional estimation, taking as input various classical estimators (e.g., $k$-NNs, kernel methods, etc.) and proposes a robustification approach considering only perturbation in the response variable. Our method whereas allows perturbations both to the covariate and response variables, which is technically more subtle because of the local conditioning problem. Within the $k$-NN DRO conditional robustification, our numerical experiments in Section 4 show substantial benefits of our local conditioning approach, especially in dealing with non-homogeneity and sharp variations in the underlying density. Our proposed framework is also relevant to the emerging stream of decision making with side information, where recent approaches rely on sample average approximation [19], decision forests [18] and probability trimmings [11].

**Notations.** For any integer $M \in \mathbb{N}_+$, we denote by $[M]$ the set $\{1, \ldots, M\}$. For any set $\mathcal{S}$, $\mathcal{M}(\mathcal{S})$ is the space of all probability measures supported on $\mathcal{S}$.

## 2 Local Conditional Estimate using Type-$\infty$ Wasserstein Ambiguity Set

We start by delineating the building blocks of our distributionally robust estimation problem (2). The nominal measure is set to the empirical distribution of the available data, $\widehat{\mathbb{P}} = N^{-1} \sum_{i \in [N]} \delta_{(\widehat{x}_i, \widehat{y}_i)}$,

where $\delta_{(\widehat{x},\widehat{y})}$ represents the Dirac distribution at $(\widehat{x},\widehat{y})$. The ambiguity set $\mathbb{B}_\rho^\infty$ is a Wasserstein ball around $\widehat{\mathbb{P}}$ that contains the true distribution $\mathbb{P}$ with high confidence.

**Definition 2.1** (Wasserstein distance). Let $\mathbb{D}$ be a metric on $\Xi$. The type-$p$ $(1 \leq p < +\infty)$ Wasserstein distance between $\mathbb{Q}_1$ and $\mathbb{Q}_2$ is defined as

$$\mathbb{W}_p(\mathbb{Q}_1,\mathbb{Q}_2) \triangleq \inf\left\{ \left(\mathbb{E}_\pi[\mathbb{D}(\xi_1,\xi_2)^p]\right)^{\frac{1}{p}} : \pi \in \Pi(\mathbb{Q}_1,\mathbb{Q}_2)\right\},$$

where $\Pi(\mathbb{Q}_1,\mathbb{Q}_2)$ is the set of all probability measures on $\Xi \times \Xi$ with marginals $\mathbb{Q}_1$ and $\mathbb{Q}_2$, respectively. The type-$\infty$ Wasserstein distance is defined as the limit of $\mathbb{W}_p$ as $p$ tends to $\infty$ and amounts to

$$\mathbb{W}_\infty(\mathbb{Q}_1,\mathbb{Q}_2) \triangleq \inf\left\{ \operatorname*{ess\,sup}_\pi \left\{\mathbb{D}(\xi_1,\xi_2) : (\xi_1,\xi_2) \in \Xi \times \Xi\right\} : \pi \in \Pi(\mathbb{Q}_1,\mathbb{Q}_2)\right\}.$$

We assume that $(X,Y)$ admits values in $\mathcal{X} \times \mathcal{Y} \subseteq \mathbb{R}^n \times \mathbb{R}^m$, and the distance $\mathbb{D}$ on $\mathcal{X} \times \mathcal{Y}$ is

$$\mathbb{D}\big((x,y),(x',y')\big) = \mathbb{D}_\mathcal{X}(x,x') + \mathbb{D}_\mathcal{Y}(y,y') \qquad \forall (x,y),(x',y') \in \mathcal{X} \times \mathcal{Y},$$

where $\mathbb{D}_\mathcal{X}$ and $\mathbb{D}_\mathcal{Y}$ are continuous metric on $\mathcal{X}$ and $\mathcal{Y}$, respectively. The joint ambiguity set $\mathbb{B}_\rho^\infty$ is now formally defined as a type-$\infty$ Wasserstein ball in the space of joint probability measures

$$\mathbb{B}_\rho^\infty \triangleq \left\{\mathbb{Q} \in \mathcal{M}(\mathcal{X} \times \mathcal{Y}) : \mathbb{W}_\infty(\mathbb{Q},\widehat{\mathbb{P}}) \leq \rho\right\}.$$

We assume further that the compact neighborhood $\mathcal{N}_\gamma(x_0)$ around $x_0$ is prescribed using the distance $\mathbb{D}_\mathcal{X}$ as $\mathcal{N}_\gamma(x_0) \triangleq \{x \in \mathcal{X} : \mathbb{D}_\mathcal{X}(x,x_0) \leq \gamma\}$, and the loss function $\ell$ is jointly continuous in $y$ and $\beta$.

To solve the estimation problem (2), we study the worst-case conditional expected loss function

$$f(\beta) \triangleq \sup_{\mathbb{Q} \in \mathbb{B}_\rho, \mathbb{Q}(X \in \mathcal{N}_\gamma(x_0))>0} \mathbb{E}_\mathbb{Q}\big[\ell(Y,\beta)|X \in \mathcal{N}_\gamma(x_0)\big],$$

which corresponds to the inner maximization problem of (2). To ensure that the value $f(\beta)$ is well-defined, we first investigate the conditions under which the above supremum problem has a non-empty feasible set. Towards this end, for any set $\mathcal{N}_\gamma(x_0) \subset \mathcal{X}$, define the quantities $\kappa_{i,\gamma}$ as

$$0 \leq \kappa_{i,\gamma} \triangleq \min_{x \in \mathcal{N}_\gamma(x_0)} \mathbb{D}_\mathcal{X}(x,\widehat{x}_i) + \inf_{y \in \mathcal{Y}} \mathbb{D}_\mathcal{Y}(y,\widehat{y}_i) \qquad \forall i \in [N]. \tag{3}$$

The value $\kappa_{i,\gamma}$ signifies the unit cost of moving a point mass from an observation $(\widehat{x}_i,\widehat{y}_i)$ to the fiber set $\mathcal{N}_\gamma(x_0) \times \mathcal{Y}$. We also define $\widehat{x}_i^p$ as the projection of $\widehat{x}_i$ onto the neighborhood $\mathcal{N}_\gamma(x_0)$, which coincides with the optimal solution in the variable $x$ of the minimization problem in (3). The next proposition asserts that $f(\beta)$ is well-defined if the radius $\rho$ is sufficiently large.

**Proposition 2.2** (Minimum radius). For any $x_0 \in \mathcal{X}$ and $\gamma \in \mathbb{R}_+$, there exists a distribution $\mathbb{Q} \in \mathbb{B}_\rho$ that satisfies $\mathbb{Q}(X \in \mathcal{N}_\gamma(x_0)) > 0$ if and only if $\rho \geq \min_{i \in [N]} \kappa_{i,\gamma}$.

We now proceed to the reformulation of $f(\beta)$. Let $\mathcal{I}$ be the index set defined as

$$\mathcal{I} \triangleq \{i \in [N] : \mathbb{D}_\mathcal{X}(x_0,\widehat{x}_i) \leq \rho + \gamma\}, \tag{4a}$$

and $\mathcal{I}$ is decomposed further into two disjoint subsets

$$\mathcal{I}_1 = \{i \in \mathcal{I} : \mathbb{D}_\mathcal{X}(x_0,\widehat{x}_i) + \rho \leq \gamma\} \quad \text{and} \quad \mathcal{I}_2 = \mathcal{I}\backslash\mathcal{I}_1. \tag{4b}$$

Intuitively speaking, $\mathcal{I}$ contains the indices of data points whose covariate $\widehat{x}_i$ is sufficiently close to $x_0$ measured by $\mathbb{D}_\mathcal{X}$, and are thus relevant to the local estimation problem. The index set $\mathcal{I}_1$ indicates the data points that lie strictly inside the neighborhood, while the set $\mathcal{I}_2$ contains those points that are on the boundary ring of width $\rho$ around the neighborhood $\mathcal{N}_\gamma(x_0)$. The value $f(\beta)$ can be efficiently computed in a quasi-closed form thanks to the following result.

**Theorem 2.3** (Worst-case conditional expected loss computation). For any $\gamma \in \mathbb{R}_+$, suppose that $\rho \geq \min_{i \in [N]} \kappa_{i,\gamma}$. For any $\beta \in \mathcal{Y}$, let $v_i^\star(\beta)$ be defined as

$$v_i^\star(\beta) \triangleq \sup_{y_i} \{\ell(y_i,\beta) : y_i \in \mathcal{Y}, \ \mathbb{D}_\mathcal{Y}(y_i,\widehat{y}_i) \leq \rho - \mathbb{D}_\mathcal{X}(\widehat{x}_i^p,\widehat{x}_i)\} \quad \forall i \in \mathcal{I}. \tag{5}$$

The worst-case conditional expected loss is equal to $f(\beta) = \left(\sum_{i\in\mathcal{I}}\alpha_i\right)^{-1}\sum_{i\in\mathcal{I}}\alpha_i v_i^\star(\beta)$, where $\alpha$ admits the value

$$\forall i\in\mathcal{I}:\quad \alpha_i = \begin{cases} 1 & \text{if } i\in\mathcal{I}_1 \text{ or } (\mathcal{I}_1=\emptyset \text{ and } v_i^\star(\beta)=\max_{j\in\mathcal{I}_2}v_j^\star(\beta)), \\ 1 & \text{if } v_i^\star(\beta) > \dfrac{\sum_{i\in\mathcal{I}_1}v_i^\star(\beta)+\sum_{j\in\mathcal{I}_2:v_j^\star(\beta)>v_i^\star(\beta)}v_j^\star(\beta)}{|\mathcal{I}_1|+|\{j\in\mathcal{I}_2:v_j^\star(\beta)>v_i^\star(\beta)\}|}, \\ 0 & \text{otherwise.}\end{cases}$$

If we possess an oracle that evaluates (5) at a complexity $\mathcal{O}$, then by Theorem 2.3, quantifying $f(\beta)$ is reduced to calculating $|\mathcal{I}|$ values of $v_i^\star(\beta)$ and then sorting these values in order to determine the value of $\alpha$. Thus, computing $f(\beta)$ takes an amount of time of order $O\big(|\mathcal{I}|(\log|\mathcal{I}|+\mathcal{O})\big)$. Moreover, $f(\beta)$ depends solely on the observations in the locality of $x_0$ whose indices belong to the index set $\mathcal{I}$, the cardinality of which can be substantially smaller than the total number of training samples $N$.

If $\ell$ is a convex function in $\beta$, then a standard result from convex analysis implies that $f$, being a pointwise supremum of convex functions, is also convex. If $\mathcal{Y}$, and hence $\beta$, is unidimensional, a golden section search algorithm can be utilized to identify the local conditional estimate $\beta^*$ that solves (2) in an amount of time of order $O\big(\log(1/\epsilon)|\mathcal{I}|(\log(|\mathcal{I}|)+\mathcal{O})\big)$, where $\epsilon>0$ is an arbitrary accuracy level. Fortunately, in the case of conditional mean and quantile estimation, we also have access to the closed form expressions of $v_i^\star(\beta)$ as long as $\mathbb{D}_\mathcal{Y}$ is an absolute distance.

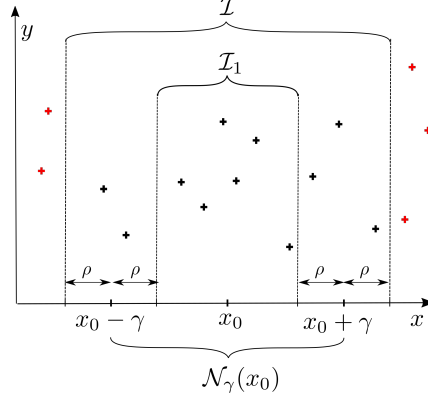

Figure 1: Illustration around the neighborhood of $x_0$ with $\rho<\gamma$. Black crosses are samples in the set $\mathcal{I}$.

**Corollary 2.4** (Value of $v_i^\star(\beta)$). *Suppose that $\mathcal{Y}=[a,b]\subseteq[-\infty,+\infty]$ and $\mathbb{D}_\mathcal{Y}(y_i,\widehat{y}_i)=|y_i-\widehat{y}_i|$.*

(i) *Conditional mean estimation: if $\ell(y,\beta)=(y-\beta)^2$, then $\forall i\in\mathcal{I}$*

$$v_i^\star(\beta)=\max\left\{(\max\{\widehat{y}_i+\rho-\mathbb{D}_\mathcal{X}(\widehat{x}_i^p,\widehat{x}_i),a\}-\beta)^2,(\min\{\widehat{y}_i+\rho-\mathbb{D}_\mathcal{X}(\widehat{x}_i^p,\widehat{x}_i),b\}-\beta)^2\right\}.$$

(ii) *Conditional quantile estimation: if $\ell(y,\beta)=\max\{-\tau(y-\beta),(1-\tau)(y-\beta)\}$, then $\forall i\in\mathcal{I}$*

$$v_i^\star(\beta)=\max\left\{-\tau(\max\{\widehat{y}_i+\rho-\mathbb{D}_\mathcal{X}(\widehat{x}_i^p,\widehat{x}_i),a\}-\beta),(1-\tau)(\min\{\widehat{y}_i+\rho-\mathbb{D}_\mathcal{X}(\widehat{x}_i^p,\widehat{x}_i),b\}-\beta)\right\}.$$

If $\mathcal{Y}$ is multidimensional, the structure of $\ell(y,\beta)$ and $\mathbb{D}_\mathcal{Y}$ might be exploited to identify tractable optimization reformulations. The next result focuses on the local conditional mean estimation.

**Proposition 2.5** (Multivariate conditional mean estimation). *Let $\mathcal{Y}=\mathbb{R}^m$ and $\ell(y,\beta)=\|y-\beta\|_2^2$.*

(i) *Suppose that $\mathbb{D}_\mathcal{Y}$ is a 2-norm on $\mathcal{Y}$, that is, $\mathbb{D}_\mathcal{Y}(y,\widehat{y})=\|y-\widehat{y}\|_2$. The distributionally robust local conditional estimation problem (2) is equivalent to the second-order cone program*

$$\begin{aligned}\min\quad & \lambda \\ \text{s.t.}\quad & \beta\in\mathbb{R}^m,\ \lambda\in\mathbb{R},\ u_i\in\mathbb{R}\ \forall i\in\mathcal{I}_1,\ u_i\in\mathbb{R}_+\ \forall i\in\mathcal{I}_2,\ t_i\in\mathbb{R}_+\ \forall i\in\mathcal{I} \\ & \textstyle\sum_{i\in\mathcal{I}}u_i\le 0,\quad t_i\ge\|\widehat{y}_i-\beta\|_2\quad\forall i\in\mathcal{I} \\ & \|[t_i+\rho-\mathbb{D}_\mathcal{X}(\widehat{x}_i^p,\widehat{x}_i)\ ;\ (1/2)(1-\lambda-u_i)]\|_2\le(1/2)(1+\lambda+u_i)\quad\forall i\in\mathcal{I}.\end{aligned}$$

(ii) *Suppose that $\mathbb{D}_\mathcal{Y}$ is a $\infty$-norm on $\mathcal{Y}$, that is, $\mathbb{D}_\mathcal{Y}(y,\widehat{y})=\|y-\widehat{y}\|_\infty$. The distributionally robust local conditional estimation problem (2) is equivalent to the second-order cone program*

$$\begin{aligned}\min\quad & \lambda \\ \text{s.t.}\quad & \beta\in\mathbb{R}^m,\ \lambda\in\mathbb{R},\ T\in\mathbb{R}_+^{|\mathcal{I}|\times m},\ u_i\in\mathbb{R}\ \forall i\in\mathcal{I}_1,\ u_i\in\mathbb{R}_+\ \forall i\in\mathcal{I}_2 \\ & \textstyle\sum_{i\in\mathcal{I}}u_i\le 0,\ ;\ \|[T_{i1}\ ;\ T_{i2}\ ;\ \cdots\ ;\ T_{im}\ ;\ \tfrac12(1-\lambda-u_i)]\|_2\le\tfrac12(1+\lambda+u_i)\ \forall i\in\mathcal{I} \\ & \left.\begin{array}{l}T_{ij}\le\widehat{y}_{ij}-\beta_j-\rho+\mathbb{D}_\mathcal{X}(\widehat{x}_i^p,\widehat{x}_i)\le T_{ij} \\ T_{ij}\le\widehat{y}_{ij}-\beta_j+\rho-\mathbb{D}_\mathcal{X}(\widehat{x}_i^p,\widehat{x}_i)\le T_{ij}\end{array}\right\}\forall(i,j)\in\mathcal{I}\times[m],\end{aligned}$$

*where $\widehat{y}_{ij}$ and $\beta_j$ are the $j$-th component of $\widehat{y}_i$ and $\beta$, respectively.*

Both optimization problems presented in Proposition 2.5 can be solved in large scale by commercial optimization solvers such as MOSEK [27]. For other multivariate conditional estimation problems, there is also a possibility of employing subgradient methods by leveraging on the next proposition.

**Proposition 2.6** (Subgradient of $f$). Suppose that $\mathbb{D}_{\mathcal{Y}}$ is coercive and $\ell(y, \cdot)$ is convex. Under the conditions of Theorem 2.3, for any $\beta \in \mathbb{R}^m$, a subgradient of the function $f$ at $\beta$ is given by $\partial f(\beta) = (\sum_{i \in \mathcal{I}} \alpha_i)^{-1} \sum_{i \in \mathcal{I}} \alpha_i \partial_\beta \ell(y_i^\star, \beta)$, where the value of $\alpha$ is as defined in Theorem 2.3 and $y_i^\star$ satisfies $y_i^\star \in \{y_i \in \mathcal{Y} : \mathbb{D}_{\mathcal{Y}}(y_i, \widehat{y}_i) \le \rho - \mathbb{D}_{\mathcal{X}}(\widehat{x}_i^p, \widehat{x}_i), \ \ell(y_i^\star, \beta) = v_i^\star(\beta)\}$ for all $i \in \mathcal{I}$.

Just as an adversarial example provides a description on how to optimally perturb a data point from the adversary's viewpoint [20, 36], the worst-case distribution provides full information on how to adversarially perturb the empirical distribution $\widehat{\mathbb{P}}$. For our distributionally robust estimator, the worst-case distribution can be obtained from the result of Theorem 2.3.

**Lemma 2.7** (Worst-case distribution). Fix an estimate $\beta \in \mathcal{Y}$. Suppose that $\rho \ge \min_{i \in [N]} \kappa_{i,\gamma}$ and let $v^\star(\beta)$ and $\alpha$ be determined as in Theorem 2.3. Moreover, let $y_i^\star$ satisfy $y_i^\star \in \{y_i \in \mathcal{Y} : \mathbb{D}_{\mathcal{Y}}(y_i, \widehat{y}_i) \le \rho - \mathbb{D}_{\mathcal{X}}(\widehat{x}_i^p, \widehat{x}_i), \ \ell(y_i^\star, \beta) = v_i^\star(\beta)\}$ for all $i \in \mathcal{I}$. Then the distribution

$$\mathbb{Q}^\star = \frac{1}{N} \left( \sum_{i \in \mathcal{I} : \alpha_i = 1} \delta_{(\widehat{x}_i^p, y_i^\star)} + \sum_{i \in \mathcal{I} : \alpha_i = 0} \delta_{(\widehat{x}_i, \widehat{y}_i)} + \sum_{i \in [N] \setminus \mathcal{I}} \delta_{(\widehat{x}_i, \widehat{y}_i)} \right)$$

satisfies $f(\beta) = \mathbb{E}_{\mathbb{Q}^\star}\big[\ell(Y, \beta) | X \in \mathcal{N}_\gamma(x_0)\big]$.

The values of $\alpha$ calculated in Theorem 2.3 are of indicative nature: $\alpha_i = 1$ if it is optimal to perturb the sample point $i$ to compute the worst-case conditional expected loss. The construction of the worst-case distribution is hence intuitive: it involves computing and sorting the values $v_i^\star(\beta)$, and then performing a greedy assignment in order to maximize the objective value.

# 3  Probabilistic Theoretical Properties

We now study the some statistical properties of our proposed estimator. Under some regularity conditions, the type-$\infty$ Wasserstein ball can be viewed as a confidence set that contains the true distribution $\mathbb{P}$ with high probability, provided that the radius $\rho$ is chosen judiciously. The value $f(\beta^\star)$ thus constitutes a generalization bound on the out-of-sample performance of the optimal conditional estimate $\beta^\star$. This idea can be formalized as follows.

**Proposition 3.1** (Finite sample guarantee). Suppose that $\mathcal{X} \times \mathcal{Y}$ is bounded, open, connected with a Lipschitz boundary. Suppose that the true probability measure $\mathbb{P}$ of $(X, Y)$ admits a density function $\nu$ satisfying $\bar{\nu}^{-1} \le \nu(x, y) \le \bar{\nu}$ for some constant $\bar{\nu} \ge 1$. For any $\gamma > 0$, if

$$\rho \ge \begin{cases} CN^{-\frac{1}{2}} \log(N)^{\frac{3}{4}} & \text{when } n + m = 2, \\ CN^{-\frac{1}{n+m}} \log(N)^{\frac{1}{n+m}} & \text{otherwise,} \end{cases}$$

where $C$ is a constant dependent on $\mathcal{X} \times \mathcal{Y}$ and $\bar{\nu}$, then for a probability of at least $1 - O(N^{-c})$, where $c > 1$ is a constant dependent on $C$, we have $\mathbb{E}_{\mathbb{P}}[\ell(Y, \beta^\star) | X \in \mathcal{N}_\gamma(x_0)] \le f(\beta^\star)$, where $\beta^\star$ is the optimal conditional estimate that solves problem (2).

We now switch gear to study the properties of our estimator in the asymptotic regime, in particular, we focus on the consistency of our estimator. The interplay between the neighborhood radius $\gamma$ and the ambiguity size $\rho$ often produces tangling effects on the asymptotic convergence of the estimate. We thus showcase two exemplary setups with either $\gamma$ or $\rho$ is zero, which interestingly produce two opposite outcomes on the consistency of the estimator. This underlines the intricacy of the problem.

**Example 3.2** (Non-consistency when $\gamma = 0$). Suppose that $\gamma = 0$, $\rho \in \mathbb{R}_{++}$ be a fixed constant, $\mathcal{Y} = \mathbb{R}$, $\ell(y, \beta) = (y - \beta)^2$, and $\mathbb{D}_{\mathcal{Y}}$ is the absolute distance. Let $\beta_N^\star$ be the optimal estimate that solves (2) dependent on $\{(\widehat{x}_i, \widehat{y}_i)\}_{i=1,\ldots,N}$. If under the true distribution $\mathbb{P}$, $X$ is independent of $Y$, $\mathbb{P}(\mathbb{D}_{\mathcal{X}}(X, x_0) \le \rho) > 0$, $\mathbb{P}(Y \ge 0) = 1$ and $\mathbb{P}(Y \ge y) > 0 \ \forall y > 0$, then with probability 1, we have $\widehat{\beta}_N \to +\infty$ while $\mathbb{E}_{\mathbb{P}}[Y | X = x_0] < \infty$.

**Example 3.3** (Consistency when $\rho = 0$). Suppose that $\rho = 0$, $\mathcal{Y} = \mathbb{R}$, $\ell(y, \beta) = (y - \beta)^2$, $\mathbb{D}_{\mathcal{X}}$ and $\mathbb{D}_{\mathcal{Y}}$ are the Euclidean distance, $k_N$ is a sequence of integer. Let $\gamma$ be the $k_N$-th smallest value of

$\mathbb{D}_{\mathcal{X}}(x_0, \widehat{x}_i)$, then $\beta_N^{\star}$ that solves (2) recovers the $k_N$-nearest neighbor regression estimator. If $k_N$ satisfies $\lim_{N\to\infty} k_N = \infty$ and $\lim_{N\to\infty} k_N/N = 0$, and $\mathbb{E}_{\mathbb{P}}[Y|X=x]$ is a continuous function of $x$, then $\lim_{N\to\infty} \beta_N^{\star} = \mathbb{E}_{\mathbb{P}}[Y|X=x_0]$ by [34, Corollary 3].

Example 3.3 suggests that if the radius $\gamma$ of the neighborhood is chosen adaptively based on the available training data, then our proposed estimator coincides with the $k$-nearest neighbor estimator, and hence consistency is inherited in a straightforward manner. The robust estimator with an ambiguity size $\rho > 0$ and an adaptive neighborhood radius $\gamma$ can thus be considered as a robustification of the $k$-nearest neighbor, which is obtained in a systematic way using the DRO framework.

It is desirable to provide a descriptive connection between the distributionally robust estimator vis-à-vis some popular statistical quantities. For the local conditional mean estimation, our estimate $\beta^{\star}$ coincides with the conditional mean of the distribution with the highest conditional variance. This insight culminates in the next proposition and bolsters the explainability of this class of estimators.

**Proposition 3.4** (Conditional mean estimate). Suppose that $\mathcal{Y} = \mathbb{R}$, $\ell(y, \beta) = (y - \beta)^2$ and $\mathbb{D}_{\mathcal{Y}}(\cdot, \widehat{y})$ is convex, coercive for any $\widehat{y}$. For any $\rho \geq \min_{i\in[N]} \kappa_{i,\gamma}$, define $\mathbb{Q}^{\star}$ as

$$\mathbb{Q}^{\star} = \arg\max_{\mathbb{Q}\in\mathbb{B}_{\rho}^{\infty}, \mathbb{Q}(X\in\mathcal{N}_{\gamma}(x_0))>0} \text{Variance}_{\mathbb{Q}}(Y|X \in \mathcal{N}_{\gamma}(x_0)),$$

then $\beta^{\star} = \mathbb{E}_{\mathbb{Q}^{\star}}[Y|X \in \mathcal{N}_{\gamma}(x_0)]$ is the optimal estimate that solves problem (2).

# 4 Numerical Experiment

In this section we compare the quality of our proposed Distributionally Robust Conditional Mean Estimator (DRCME) to $k$-nearest neighbour ($k$-NN), Nadaraya-Watson (N-W), and Nadaraya-Epanechnikov (N-E) estimators, together with the robust $k$-NN approach in [2] (BertEtAl) using a synthetic and the MNIST datasets. Codes are available at `https://github.com/nvietanh/DRCME`.

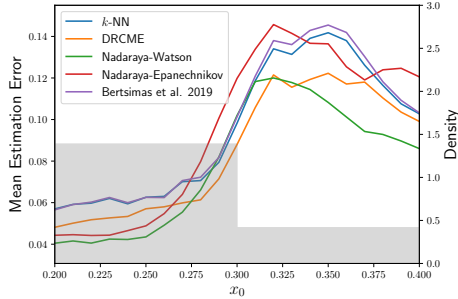
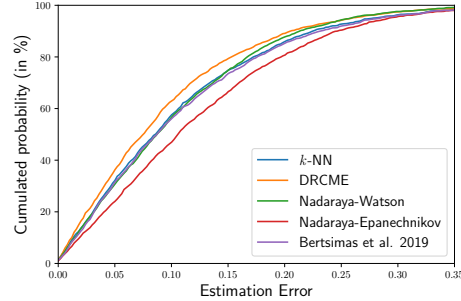

Figure 2: Comparison of the mean absolute errors of conditional mean estimators for synthetic data. The gray shade shows the density of $X$.

Figure 3: Comparison of the distributions of absolute estimation errors of conditional mean estimators for synthetic data.

## 4.1 Conditional Mean Estimation With Synthetic Data

In this section, we conducted 500 independent experiments where the training set contains $N = 100$ i.i.d. samples of $(X, Y)$ in each experiment. The marginal distribution of $X$ has piecewise constant density function $p(x)$, which is chosen as $p(x) = 100/72$ if $x \in [0, 0.3] \cup [0.7, 1]$ and $p(x) = 30/72$ if $x \in (0.3, 0.7)$. Given $X$, the distribution of $Y$ is determined by $Y = f(X) + \varepsilon$, where $f = \sin(10 \cdot x)$ and $\varepsilon$ is i.i.d. Gaussian noise independent of $X$ with mean 0 and variance 0.01. The conditional mean estimation problem is challenging when $x_0$ is close to the jump points of the density function $p(x)$, that is at $x_0 = 0.3$ or $x_0 = 0.7$, because the data are gathered unequally in the neighborhoods. Thus, to test the robustness of all the estimators, we employ all the five estimators to estimate the conditional mean $\mathbb{E}_{\mathbb{P}}[Y|X=x_0]$, for $x_0 = 0.2, 0.21, \ldots, 0.4$ around the jump point $x_0 = 0.3$. We select $\mathbb{D}_{\mathcal{X}}(x, x') = |x - x'|$ and $\mathbb{D}_{\mathcal{Y}}(y, y') = |y - y'|$. The hyperparameters of all the estimators, whose range and selection are given in Appendix A, are chosen by leave-one-out cross validation.

Figure 2 displays the average of the mean estimation errors taken over 500 independent runs for different values $x_0 \in [0.2, 0.4]$. One can observe from the figure that DRCME uniformly outperforms

| Method | H.P. | $N$=50 | $N$=100 | $N$=500 |
|---|---|---|---|---|
| $k$-NN | $k$ | 3 | 4 | 4 |
| N-W | $h$ | 0.022 | 0.019 | 0.015 |
| N-E | $h$ | 0.087 | 0.078 | 0.068 |
| BertEtAl | $k$ | 3 | 4 | 5 |
| | $\rho$ | 0.712 | 1.313 | 1.313 |
| DRCME | $\gamma$ | $h_{1.3}^{\gamma}(\cdot)$ | $h_{1.3}^{\gamma}(\cdot)$ | $h_{1.6}^{\gamma}(\cdot)$ |
| | $\rho$ | $0.13\gamma$ | $0.13\gamma$ | $0.06\gamma$ |
| | $\theta$ | 0.004 | 0.002 | 0.001 |

Table 1: Median of hyper-parameters (H.P.) obtained with cross-validation.

| Method | $N$=50 | $N$=100 | $N$=500 |
|---|---|---|---|
| $k$-NN | $24 \pm 2$ | $33 \pm 2$ | $60 \pm 1$ |
| N-W | $30 \pm 2$ | $38 \pm 2$ | $65 \pm 1$ |
| N-E | $26 \pm 1$ | $32 \pm 1$ | $50 \pm 1$ |
| BertEtAl | $29 \pm 2$ | $41 \pm 2$ | $67 \pm 1$ |
| DRCME | $36 \pm 2$ | $46 \pm 1$ | $71 \pm 1$ |

Table 2: Comparison of expected out-of-sample classification accuracy (in % with 90% confidence intervals) from rounded estimates.

$k$-NN, BertEtAl for all $x_0$ of interest. When compared with N-W and N-E, we remark that DRCME is the most accurate estimator around the jump point of $p(x)$. As $x_0$ moves away from the location 0.3, the performance of DRCME decays and becomes slightly worse than N-W as $x_0$ goes far from the jump point. Figure 3 presents the cumulative distribution of the estimation errors when $x_0 \in [0.28, 0.32]$. The empirical error distribution of DRCME is stochastically smaller than that of other estimators, which reinforces that DRCME outperforms around the jump point in a strong sense.

## 4.2 Digit Estimation With MNIST Database

In this section, we compare the quality of the estimators on a digit estimation problem using the MNIST database [23]. While to this date most studies have focused on out-of-sample classification performances for this dataset, here we shift our attention to the task of estimation of digits as **cardinal** quantities and are especially interested in performance at a low-data regime. Treating the labels as cardinal quantities allows us to assess the distinctive features of DRCME in its most simplistic form (i.e. univariate conditional mean estimation of a real random variable). Mean estimation might in fact be more relevant than classification when trying to recognize handwritten measurements where confusing a 0 with a 6 is more damaging than with a 3.

We executed 100 experiments where training and test sets were randomly drawn without replacement from the 60,000 training examples of this dataset. Training set sizes were $N = 50, 100$, or 500 while test sets' size remained at 100. Each $(x, y)$ pair is composed of the normalized vector, in $\mathbb{R}^{28^2}$ of grayscale intensities normalized so that $\|x\|_1 = 1$. For simplicity, we let $\mathbb{D}_{\mathcal{X}}(x, \widehat{x}) = \|x - \widehat{x}\|_2$ and $\mathbb{D}_{\mathcal{Y}}(y, \widehat{y}) = \theta|y - \widehat{y}|$. In each experiment, the hyper-parameters of all four methods were chosen based on a leave-one-out cross validation process. In the case of DRCME, we adapt the radius of the neighborhood $\gamma$ and $\rho$ locally at $x_0$ to account for the non-uniform density of $X$.[1] Table 1 presents the median choice of hyper parameters for each estimator.

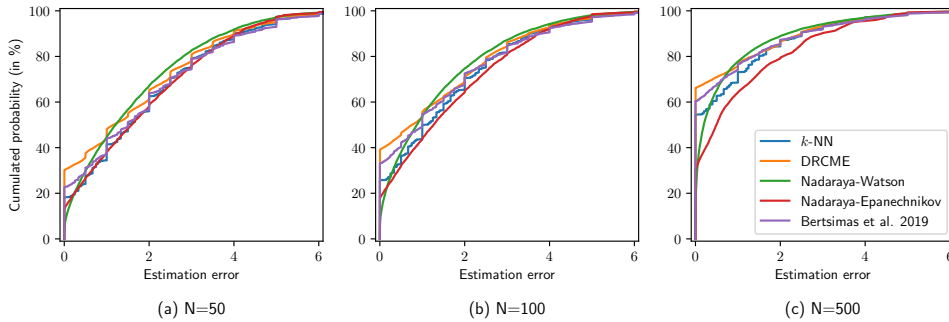

(a) N=50     (b) N=100     (c) N=500

Figure 4: Comparison of the distributions of out-of-sample absolute estimation errors of conditional mean estimators for the MNIST database under different training set sizes.

Figure 4 presents the out-of-sample estimation error distribution of all four conditional estimators. One can quickly remark that the DRCME outperforms BertEtAl, $k$-NN, and N-E estimators, especially

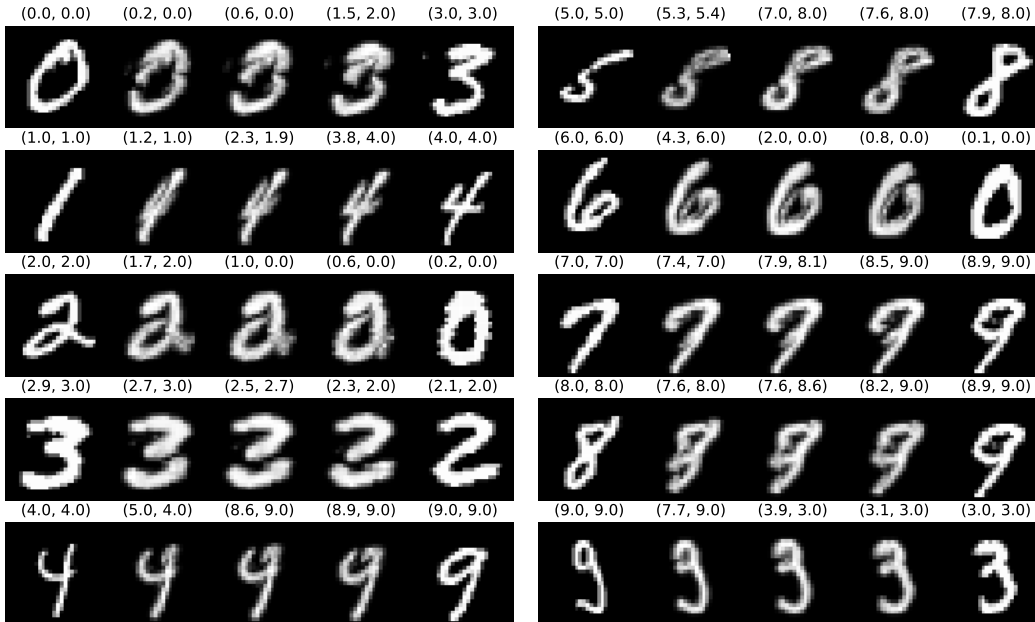

Figure 5: Comparison of estimations from N-W and DRCME on entropic regularized Wasserstein barycenters of pairs of images from the training set. Estimations are presented above each image in the format "(N-W, DRCME)".

for low-data regime. In particular, for all three training set sizes, the distribution of error for DRCME stochastically dominates the three other distributions. In particular, one even notices in (c) that DRCME has the largest chance of reaching an exact estimation: 66% compared to 60%, 55%, 30%, and 8% for the other estimators. This explains why DRCME is also the most accurate estimator when rounding it to the nearest integer as reported in Table 2: with a margin greater than 4% from all estimators across all $N$'s. It is worth noting that while N-W does not produce high accuracy estimate, it however has less chances of producing estimation with large errors. This is also apparent when comparing the expected type-$p$ deviation of the estimation error, i.e. $(\mathbb{E}[|y - \hat{y}|^p])^{1/p}$, for each estimator. Specifically, N-W slightly outperforms DRCME for deviation metrics of type $p \geq 1$, e.g. with a root mean square error of 1.32 compared to 1.41 when $N = 500$. On the other hand, DRCME significantly outperforms N-W when $p < 1$ where high precision estimators are encouraged. We refer the reader to Appendix A for further details.

Finally, we report on an experiment that challenges the capacity of both N-W and DRCME estimators to be resilient to adversarial corruption of the test images. This is done by exposing the two estimators to images from the training set ($N = 100$) that have been corrupted in a way that makes them resemble the closest differently-labeled image in the set.[2] Figure 5 presents several visual examples of the progressively corrupted images and the resulting N-W and DRCME estimations. Overall, one quickly notices how the estimation produced by DRCME is less sensitive to such attacks, "sticking" to the original label until there is substantial evidence of a new label. More examples are in Appendix A.

## Broader Impact

Our paper contributes theoretical insights at the intersection of statistics and optimization, with potential applications in diverse areas of machine learning. In particular, our proposed estimator can be used in almost all applications in which the non-parametric conditional estimators (including $k$-nearest neighbors and kernel estimators) are currently utilized, including regression and classification tasks with potential impact in health sciences, economics, business, finance, climate, various engineering areas, logistics, risk analysis, etc. Using ideas from the distributionally robust optimization framework, we propose a principled and systematic way to obtain a robustification of the popular $k$-nearest

neighbors. At a methodological level, we contribute a novel paradigm that can be used to enhance robustness of conditional statistical estimation against model misspecification and adversarial attacks.

Because our paper provides novel techniques for conditional estimation in the context structured and contaminated data, we believe that we have the potential of enabling more applications in which data sets are pulled together from different sources (e.g., for prediction of health care policy evaluations in which information from different environments needs to be put together to mitigate the lack of data given the need for quick decision making under time constraints; for online advertisement recommendation system in which the behaviors of many customers are employed to predict the behavior of incoming customers conditional on their profile). In addition, the results in this paper are a part of the thesis work of a Ph.D. student, thus promoting the training for highly qualified personnel.

**Acknowledgments.** Material in this paper is based upon work supported by the Air Force Office of Scientific Research under award number FA9550-20-1-0397. Additional support is gratefully acknowledged from NSF grants 1915967, 1820942, 1838676, NSERC grant RGPIN-2016-05208, and from the China Merchant Bank. Finally, this research was enabled in part by support provided by Compute Canada.

## Footnotes

[1]Specifically, we let $\gamma = h_i^{\gamma}(x_0) := \kappa_{[\lfloor i \rfloor], 0} + (i - \lfloor i \rfloor)(\kappa_{[\lceil i \rceil], 0} - \kappa_{[\lfloor i \rfloor], 0})$, where $[j]$ refers to the $j$-th smallest element while $\lfloor \cdot \rfloor$ and $\lceil \cdot \rceil$ refer to the floor and ceiling operations, i.e., the radius is set to the linear interpolation between the distance of the $\lfloor i \rfloor$-th and $\lfloor i \rfloor + 1$-th closest members of the training set to $x_0$. We further let $\rho$ be proportional to $\gamma$. This lets DRCME reduce to $k$-NN when $\gamma = h_k^{\gamma}(x_0)$, $\rho = 0$, and $\theta = 1$.

[2]Implementation wise, we exploit the Python Optimal Transport toolbox [12] to compute different entropic regularized Wasserstein barycenters of the two normalized images treated as distributions.

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
