[Supplementary Material · neurips_conditional_appendix.pdf]

# Appendix
# Distributionally Robust Local Non-parametric Conditional Estimation

## 1    A    Additional Experiment Results

## 2    A.1    Conditional Mean Estimation With Synthetic Data

3  We report in Figure A.1 the plot of mean estimation errors versus $x_0$ for different training set sizes
4  $N = 50, 100, 200$. In Figure A.2 we present the plot of the distribution of absolute estimation errors
5  for $x_0 \in [0.28, 0.32]$. For comparison, we also include the results of training set size $N = 100$ that
6  are already reported in Figure 2 and 3. We remark that the estimation error of all the estimators
7  becomes smaller when training set size is larger, and DRCME has best estimation performance
8  among all the estimators around the jump point $x = 0.3$ for all different training set sizes.

9  We report the hyper-parameters selected by cross-validation in Table A.1.

Figure A.1: Comparison of the mean absolute errors of conditional mean estimators for synthetic data under different training set sizes. The gray shade shows the density of $X$.

## 10    A.2    Digit Estimation With MNIST Database

11  The distinction between N-W and DRCME is also apparent in Figure A.3 which presents the
12  normalized expected type-$p$ deviation of the estimation error for each estimator, i.e. $\sqrt{2/p}(\mathbb{E}[|y -$
13  $\hat{y}|^p])^{1/p}$. Specifically, N-W slightly outperforms DRCME for deviation metrics of type $p \geq 1$, e.g.
14  with a root mean square error of 1.34 compared to 1.45 when $N = 500$. On the other hand, DRCME
15  significantly outperforms N-W when $p < 1$ where high precision estimators are encouraged.

(a) $N = 50$  (b) $N = 100$  (c) $N = 200$

Figure A.2: Comparison of the distributions of absolute estimation errors of conditional mean estimators for synthetic data under different training set sizes.

| Method | H.P. | $N$=50 | $N$=100 | $N$=200 |
|---|---|---|---|---|
| $k$-NN | $k$ | 1 | 3 | 5 |
| N-W | $h$ | 0.026 | 0.019 | 0.018 |
| N-E | $h$ | 0.078 | 0.055 | 0.038 |
| BertEtAl | $k$ | 1 | 3 | 5 |
| | $\rho$ | 0.063 | 0.016 | 0.000 |
| | $\gamma$ | $h_1^\gamma(\cdot)$ | $h_2^\gamma(\cdot)$ | $h_3^\gamma(\cdot)$ |
| DRCME | $\rho$ | $0.031\gamma$ | $0.063\gamma$ | $0.063\gamma$ |

Table A.1: Median of hyper-parameters (H.P.) for synthetic data experiment obtained with cross-validation.

(a)  N=50  (b)  N=100  (c)  N=500

Figure A.3: Comparison of normalized expected type-$p$ deviation of the out-of-sample error of four non-parametric conditional mean estimation methods for the MNIST database under different training set sizes. E.g., at $p = 2$ is presented the root-mean square error.

We also include in Figure A.4 some additional examples of labels from DRCME and N-W. On the other hand, Figure A.5 compares the labels from DRCME and BertEtAl .

## 18  B  Proofs

19  This section contains the proofs of all technical results presented in the main paper.

### 20  B.1  Proofs of Section 2

21  *Proof of Proposition 2.2.* Using the definition of the type-$\infty$ Wasserstein distance, we can re-express
22  the ambiguity set $\mathbb{B}_\rho^\infty$ as

$$\mathbb{B}_\rho^\infty = \left\{ \mathbb{Q} \in \mathcal{M}(\mathcal{X} \times \mathcal{Y}) : \begin{array}{l} \exists \pi \in \Pi(\mathbb{Q}, \widehat{\mathbb{P}}) \text{ such that} \\ \operatorname*{ess\,sup}_\pi \{ \mathbb{D}_\mathcal{X}(x, x') + \mathbb{D}_\mathcal{Y}(y, y') \} \le \rho \end{array} \right\}$$

$$= \left\{ \mathbb{Q} \in \mathcal{M}(\mathcal{X} \times \mathcal{Y}) : \begin{array}{l} \exists \pi_i \in \mathcal{M}(\mathcal{X} \times \mathcal{Y}) \; \forall i \in [N] \text{ such that } \mathbb{Q} = \frac{1}{N} \sum_{i \in [N]} \pi_i \\ \operatorname*{ess\,sup}_{\frac{1}{N} \sum_{i \in [N]} \pi_i \otimes \delta_{(\widehat{x}_i, \widehat{y}_i)}} \{ \mathbb{D}_\mathcal{X}(x, x') + \mathbb{D}_\mathcal{Y}(y, y') \} \le \rho \end{array} \right\},$$

23  where in the second equality we exploit the fact that $\widehat{\mathbb{P}}$ is an empirical measure and thus any joint
24  probability measure $\pi \in \Pi(\mathbb{Q}, \widehat{\mathbb{P}})$ can be written as $\pi = N^{-1} \sum_{i \in [N]} \pi_i \otimes \delta_{(\widehat{x}_i, \widehat{y}_i)}$, where each $\pi_i$ is
25  a probability measure supported on $\mathcal{X} \times \mathcal{Y}$. The last constraint can now be written as

$$\mathbb{D}_\mathcal{X}(x, \widehat{x}_i) + \mathbb{D}_\mathcal{Y}(y, \widehat{y}_i) \le \rho \quad \forall (x, y) \in \operatorname{supp}(\pi_i) \quad \forall i \in [N],$$

26  where $\operatorname{supp}(\pi_i)$ denotes the support of the probability measure $\pi_i$ [1, Page 441]. We thus have

$$\mathbb{B}_\rho^\infty = \left\{ \mathbb{Q} \in \mathcal{M}(\mathcal{X} \times \mathcal{Y}) : \begin{array}{l} \exists \pi_i \in \mathcal{M}(\mathcal{X} \times \mathcal{Y}) \; \forall i \in [N] \text{ such that } \mathbb{Q} = \frac{1}{N} \sum_{i \in [N]} \pi_i \\ \mathbb{D}_\mathcal{X}(x, \widehat{x}_i) + \mathbb{D}_\mathcal{Y}(y, \widehat{y}_i) \le \rho \quad \forall (x, y) \in \operatorname{supp}(\pi_i) \quad \forall i \in [N] \end{array} \right\}.$$

27  Suppose that $\rho < \min_{i \in [N]} \kappa_{i,\gamma}$, then this implies by the last constraint of the feasible set that
28  $\pi_i(\mathcal{N}_\gamma(x_0) \times \mathcal{Y}) = 0$ for all $i \in [N]$. As a consequence, any $\mathbb{Q} \in \mathbb{B}_\rho^\infty$ should satisfy

$$\mathbb{Q}(X \in \mathcal{N}_\gamma(x_0)) = \sum_{i \in [N]} \pi_i(\mathcal{N}_\gamma(x_0) \times \mathcal{Y}) = 0.$$

29  Hence $\mathbb{B}_\rho^\infty \cap \{ \mathbb{Q} \in \mathcal{M}(\mathcal{X} \times \mathcal{Y}) : \mathbb{Q}(X \in \mathcal{N}_\gamma(x_0)) > 0 \} = \emptyset$.

30  Suppose on the contrary that $\rho \ge \min_{i \in [N]} \kappa_{i,\gamma}$. Let $i^\star = \arg\min_{i \in [N]} \kappa_{i,\gamma}$, and consider the
31  following set of probability measures

$$\forall i \in [N] : \qquad \pi_i = \begin{cases} \delta_{(\widehat{x}_i^p, \widehat{y}_i)} & \text{if } i = i^\star, \\ \delta_{(\widehat{x}_i, \widehat{y}_i)} & \text{otherwise}, \end{cases}$$

32  and set $\mathbb{Q} = \frac{1}{N} \sum_{i \in [N]} \pi_i$. It is easy to verify that $\mathbb{Q} \in \mathbb{B}_\rho^\infty$, and that

$$\mathbb{Q}(X \in \mathcal{N}_\gamma(x_0)) \ge \frac{1}{N} \pi_{i^\star}(X \in \mathcal{N}_\gamma(x_0)) = \frac{1}{N} > 0.$$

33  This observation completes the proof. $\qquad\qquad\square$

34  The proof of Theorem 2.3 relies on the following result.

35  **Lemma B.1** (Optimal solution of a fractional linear program). Let $d$ be an strictly positive integer.
36  The linear fractional program

$$\min \left\{ \frac{c + \sum_{i=1}^K v_i \alpha_i}{d + \sum_{i=1}^K \alpha_i} : \alpha \in [0, 1]^K \right\}$$

37  admits the optimal solution

$$\forall i \in [K] : \qquad \alpha_i^\star = \begin{cases} 1 & \text{if } v_i > \dfrac{c + \sum_{j : v_j > v_i} v_j}{d + |\{ j : v_j > v_i \}|}, \\ 0 & \text{otherwise}. \end{cases}$$

38 *Proof of Lemma B.1.* Without loss of generality assume that $v_i$ are ordered decreasingly. Because
39 the objective function is pseudolinear, the optimal solution is at some binary vertex [6, Lemma 3.3].
40 Consider the equivalent problem

$$\max_{k,\alpha} \left\{ \frac{c + \sum_{i=1}^{K} v_i \alpha_i}{d + \sum_{i=1}^{K} \alpha_i} : \alpha \in \{0,1\}^K, \ \sum_{i=1}^{K} \alpha_i = k, \ k \in [K] \right\}.$$

41 For any value $k \in [K]$, the corresponding optimal value of $\alpha$ dependent on $k$ is

$$\alpha_i^\star(k) = \begin{cases} 1 & \text{if } i \le k, \\ 0 & \text{otherwise,} \end{cases}$$

42 where we exploit the fact that $v_i$ are ordered decreasingly. The above optimization problem can be
43 simplified to

$$\max_k \left\{ \frac{c + \sum_{i=1}^{k} v_i}{d + k} : k \in [K] \right\}. \tag{A.1}$$

44 Now we need to show that the objective function $g(k) \triangleq (c + \sum_{i=1}^{k} v_i)/(d + k)$ becomes non-
45 increasing once it starts decreasing. Indeed, the incremental improvement in the objective value
46 of (A.1) at $k$ can be written as

$$\begin{aligned} \Delta_g(k) = g(k+1) - g(k) &= \frac{c + \sum_{i=1}^{k+1} v_i}{d+k+1} - g(k) \\ &= \frac{(d+k)g(k) + v_{k+1}}{d+k+1} - g(k) \\ &= \frac{v_{k+1} - g(k)}{d+k+1}. \end{aligned}$$

47 If $\Delta_g(k) < 0$, this implies that $v_{k+1} < g(k)$. We also know that $v_{k+2} \le v_{k+1}$. So we can show that:

$$\begin{aligned} \Delta_g(k+1) = g(k+2) - g(k+1) &= \frac{v_{k+2} - g(k+1)}{d+k+2} \\ &= \frac{(d+k+1)v_{k+2} - (d+k)g(k) - v_{k+1}}{(d+k+2)(d+k+1)} \\ &\le \frac{(d+k+1)v_{k+1} - (d+k)g(k) - v_{k+1}}{(d+k+2)(d+k+1)} \\ &= \frac{(d+k)(v_{k+1} - g(k))}{(d+k+2)(d+k+1)} < 0. \end{aligned}$$

48 Moreover, the above line of arguments also reveals that if $v_{k+2} = v_{k+1}$ then both $\Delta_g(k)$ and
49 $\Delta_g(k+1)$ have the same sign. Thus, the value $k^\star$ that maximizes (A.1) is also the solution of

$$\max\{k : \Delta_g(k-1) \ge 0\}.$$

50 Leveraging on the formula of $\alpha_i^\star(k)$, the solution $\alpha^\star$ of the original fractional linear program has the
51 form

$$\forall i: \qquad \alpha_i^\star = \begin{cases} 1 & \text{if } v_i > \dfrac{c + \sum_{j:j<i} v_i}{d + |\{j : j < i\}|}, \\ 0 & \text{otherwise,} \end{cases}$$

$$= \begin{cases} 1 & \text{if } v_i > \dfrac{c + \sum_{j:v_j>v_i} v_j}{d + |\{j : v_j > v_i\}|}, \\ 0 & \text{otherwise,} \end{cases}$$

52 where the second equality comes from the ordering of $v_i$. This observation completes the proof. $\square$

53 *Proof of Theorem 2.3.* A conditional measure $\mu_0$ of $Y$ given $X \in \mathcal{N}_\gamma(x_0)$ induced by a probability
54 measure $\mathbb{Q}$ satisfying $\mathbb{Q}(X \in \mathcal{N}_\gamma(x_0)) > 0$ can be written as

$$\mathbb{Q}(\mathcal{N}_\gamma(x_0) \times A) = \mu_0(A)\mathbb{Q}(\mathcal{N}_\gamma(x_0) \times \mathcal{Y}) \qquad \forall A \subseteq \mathcal{Y} \text{ measurable.}$$

55   One can rewrite the worst-case conditional expected loss $f(\beta)$ as

$$f(\beta) = \begin{cases} \sup & \int_{\mathcal{Y}} \ell(y,\beta)\,\mu_0(\mathrm{d}y) \\ \text{s.\,t.} & \mathbb{Q} \in \mathbb{B}_\rho^\infty,\ \mathbb{Q}(\mathcal{N}_\gamma(x_0) \times \mathcal{Y}) > 0 \\ & \mathbb{Q}(\mathcal{N}_\gamma(x_0) \times A) = \mu_0(A)\mathbb{Q}(\mathcal{N}_\gamma(x_0) \times \mathcal{Y}) \qquad \forall A \subseteq \mathcal{Y} \text{ measurable.} \end{cases}$$

56   By decomposing the measure $\mathbb{Q}$ using the set of probability measures $\pi_i$ and exploiting the definition
57   of the type-$\infty$ Wasserstein distance as in the proof of Proposition 2.2, we have

$$f(\beta) = \begin{cases} \sup & \int_{\mathcal{Y}} \ell(y,\beta)\,\mu_0(\mathrm{d}y) \\ \text{s.\,t.} & \mu_0 \in \mathcal{M}(\mathcal{Y}),\ \pi_i \in \mathcal{M}(\mathcal{X} \times \mathcal{Y}) \quad \forall i \in [N] \\ & \sum_{i \in [N]} \pi_i(\mathcal{N}_\gamma(x_0) \times \mathcal{Y}) > 0 \\ & \sum_{i \in [N]} \pi_i(\mathcal{N}_\gamma(x_0) \times A) = \mu_0(A) \sum_{i \in [N]} \pi_i(\mathcal{N}_\gamma(x_0) \times \mathcal{Y}) \quad \forall A \subseteq \mathcal{Y} \text{ measurable} \\ & \mathbb{D}_{\mathcal{X}}(x,\widehat{x}_i) + \mathbb{D}_{\mathcal{Y}}(y,\widehat{y}_i) \le \rho \quad \forall (x,y) \in \operatorname{supp}(\pi_i) \qquad \forall i \in [N]. \end{cases}$$

58   For any set of feasible solutions $\{\pi_i\}_{i \in [N]}$, we have $\sum_{i \in [N]} \pi_i(\mathcal{N}_\gamma(x_0) \times \mathcal{Y}) > 0$. We can thus
59   re-express $\mu_0(A)$ for any Borel measurable set $A \subseteq \mathcal{Y}$ as

$$\mu_0(A) = \frac{\sum_{i \in [N]} \pi_i(\mathcal{N}_\gamma(x_0) \times A)}{\sum_{i \in [N]} \pi_i(\mathcal{N}_\gamma(x_0) \times \mathcal{Y})} \quad \forall A \subseteq \mathcal{Y} \text{ measurable.}$$

60   Thus, we can eliminate the variables $\mu_0$ from the above optimization problem to obtain the equivalent
61   representation

$$f(\beta) = \begin{cases} \sup & \frac{1}{\sum_{i \in [N]} \pi_i(\mathcal{N}_\gamma(x_0) \times \mathcal{Y})} \sum_{i \in [N]} \int_{\mathcal{Y}} \ell(y,\beta)\,\pi_i(\mathcal{N}_\gamma(x_0) \times \mathrm{d}y) \\ \text{s.\,t.} & \pi_i \in \mathcal{M}(\mathcal{X} \times \mathcal{Y}) \hspace{4cm} \forall i \in [N] \\ & \mathbb{D}_{\mathcal{X}}(x,\widehat{x}_i) + \mathbb{D}_{\mathcal{Y}}(y,\widehat{y}_i) \le \rho \quad \forall (x,y) \in \operatorname{supp}(\pi_i) \quad \forall i \in [N] \\ & \sum_{i \in [N]} \pi_i(\mathcal{N}_\gamma(x_0) \times \mathcal{Y}) > 0. \end{cases} \tag{A.2}$$

62   We now show that problem (A.2) now can be written as

$$f(\beta) = \begin{cases} \sup & \frac{1}{\sum_{i \in [N]} \alpha_i} \sum_{i \in [N]} \alpha_i v_i^\star(\beta) \\ \text{s.\,t.} & \alpha \in [0,1]^N \\ & \alpha_i = 1 \text{ if } \mathbb{D}_{\mathcal{X}}(x_0,\widehat{x}_i) + \rho \le \gamma \\ & \alpha_i = 0 \text{ if } \mathbb{D}_{\mathcal{X}}(x_0,\widehat{x}_i) > \rho + \gamma \\ & \sum_{i \in [N]} \alpha_i > 0, \end{cases} \tag{A.3}$$

63   where the value $v_i^\star(\beta)$ is calculated as

$$v_i^\star(\beta) = \sup\left\{\ell(y_i,\beta)\ :\ y_i \in \mathcal{Y},\ \mathbb{D}_{\mathcal{Y}}(y_i,\widehat{y}_i) \le \rho - \mathbb{D}_{\mathcal{X}}(\widehat{x}_i^p,\widehat{x}_i)\right\}.$$

64   The equivalence between the supremum problems (A.2) and (A.3) can be shown in two steps. First,
65   for (A.2) $\le$ (A.3), given any feasible solution of (A.2), one can construct a feasible solution of (A.3)
66   using $\alpha_i = \pi_i(\mathcal{N}_\gamma(x_0) \times \mathcal{Y})$. For this candidate we have

$$\frac{\sum_{i \in [N]} \int_{\mathcal{Y}} \ell(y,\beta)\,\pi_i(\mathcal{N}_\gamma(x_0) \times \mathrm{d}y)}{\sum_{i \in [N]} \pi_i(\mathcal{N}_\gamma(x_0) \times \mathcal{Y})} \le \frac{\sum_{i \in [N]} \alpha_i \ell(y_i^\star,\beta)}{\sum_{i \in [N]} \alpha_i}.$$

67   Alternatively, given a feasible solution for (A.3), one can construct the following feasible solution for
68   (A.2): for any $\epsilon > 0$, let $y_i^\epsilon \in \mathcal{Y}$ be such that $\mathbb{D}_{\mathcal{Y}}(y_i^\epsilon,\widehat{y}_i) \le \rho - \mathbb{D}_{\mathcal{X}}(x_0,\widehat{x}_i)$ and $\ell(y_i^\epsilon,\beta) \ge v_i^\star(\beta) - \epsilon$,
69   and let

$$\forall i \in [N]:\quad \pi_i^\epsilon = \begin{cases} \delta_{(\widehat{x}_i^p,y_i^\epsilon)} & \text{if } \mathbb{D}_{\mathcal{X}}(x_0,\widehat{x}_i) + \rho \le \gamma, \\ \alpha_i \delta_{(\widehat{x}_i^p,y_i^\epsilon)} + (1-\alpha_i)\delta_{(x_i^r,\widehat{y}_i)} & \text{if } \mathbb{D}_{\mathcal{X}}(x_0,\widehat{x}_i) > \rho + \gamma, \\ \delta_{(\widehat{x}_i,\widehat{y}_i)} & \text{otherwise,} \end{cases}$$

where $x_i^r$ is any point such that $\mathbb{D}_{\mathcal{X}}(x_i^r, \widehat{x}_i) \leq \rho$ and $x_i^r \notin \mathcal{N}_\gamma(x_0)$. Again, this candidate is feasible in (A.2) and we have that

$$
\begin{aligned}
f(\beta) &\geq \sup_{\epsilon>0} \frac{\sum_{i\in[N]} \int_{\mathcal{Y}} \ell(y,\beta)\, \pi_i^\epsilon(\mathcal{N}_\gamma(x_0) \times \mathrm{d}y)}{\sum_{i\in[N]} \pi_i^\epsilon(\mathcal{N}_\gamma(x_0) \times \mathcal{Y})} \\
&\geq \sup_{\epsilon>0} \frac{\sum_{i\in[N]} \alpha_i(\ell(y_i^\star,\beta) - \epsilon)}{\sum_{i\in[N]} \alpha_i} \\
&= \frac{\sum_{i\in[N]} \alpha_i \ell(y_i^\star,\beta)}{\sum_{i\in[N]} \alpha_i} = \frac{\sum_{i\in[N]} \alpha_i v_i^\star(\beta)}{\sum_{i\in[N]} \alpha_i}
\end{aligned}
$$

Let $\mathcal{I}$ and $\mathcal{I}_1$ be the index sets defined as in (4a)-(4b), the value $f(\beta)$ is equal to the optimal value of a fractional linear program

$$
f(\beta) = \max \left\{ \frac{\sum_{i\in\mathcal{I}} v_i^\star(\beta)\alpha_i}{\sum_{i\in\mathcal{I}} \alpha_i} : \alpha \in [0,1]^N,\ \alpha_i = 1\ \forall i \in \mathcal{I}_1,\ \sum_{i\in\mathcal{I}} \alpha_i > 0 \right\} \tag{A.4a}
$$

$$
= \max \left\{ \frac{\sum_{i\in\mathcal{I}_1} v_i^\star(\beta) + \sum_{i\in\mathcal{I}_2} v_i^\star(\beta)\alpha_i}{|\mathcal{I}_1| + \sum_{i\in\mathcal{I}_2} \alpha_i} : \alpha \in [0,1]^N,\ \alpha_i = 1\ \forall i \in \mathcal{I}_1,\ |\mathcal{I}_1| + \sum_{i\in\mathcal{I}_2} \alpha_i > 0 \right\}. \tag{A.4b}
$$

Notice that the objective function and the constraints of (A.4b) depend only on $\alpha_i$ for $i \in \mathcal{I}$. Suppose that $\mathcal{I}_1 \neq \emptyset$, Lemma B.1 indicates that the optimal solution $\alpha^\star$ that solves (A.4b) is

$$
\forall i \in \mathcal{I}: \quad \alpha_i^\star = \begin{cases} 1 & \text{if } i \in \mathcal{I}_1, \\ 1 & \text{if } v_i^\star(\beta) > \dfrac{\sum_{i\in\mathcal{I}_1} v_i^\star(\beta) + \sum_{j:v_j^\star(\beta)>v_i^\star(\beta)} v_j^\star(\beta)}{|\mathcal{I}_1| + |\{j : v_j^\star(\beta) > v_i^\star(\beta)\}|}, \\ 0 & \text{otherwise.} \end{cases} \tag{A.5}
$$

Suppose that $\mathcal{I}_1 = \emptyset$, then the optimal solution of problem (A.4b) is

$$
\forall i \in \mathcal{I}: \quad \alpha_i^\star = \begin{cases} 1 & \text{if } v_i^\star(\beta) \geq \max_{j\in\mathcal{I}_2} v_j^\star(\beta), \\ 0 & \text{otherwise.} \end{cases}
$$

Combining the above two cases, we can rewrite the optimal value of $\alpha$ that solves (A.4b) as in the statement of the theorem. This completes the proof. $\qquad\square$

*Proof of Corollary 2.4.* Because $\mathbb{D}_{\mathcal{Y}}$ is an absolute distance, we have

$$
\{y_i \in \mathcal{Y} : |y_i - \widehat{y}_i| \leq \rho - \mathbb{D}_{\mathcal{X}}(\widehat{x}_i^p, \widehat{x}_i)\} = [\max\{a, \widehat{y}_i - \rho + \mathbb{D}_{\mathcal{X}}(\widehat{x}_i^p, \widehat{x}_i)\}, \min\{b, \widehat{y}_i + \rho - \mathbb{D}_{\mathcal{X}}(\widehat{x}_i^p, \widehat{x}_i)\}],
$$

where the equality follows from $\mathcal{Y} = [a, b]$. Because both the $\|\cdot\|_2^2$ and the quantile loss functions are convex, the value $v_i^\star(\beta)$ is thus attained at the extreme points of the interval. Calculating the value of $\ell(\cdot, \beta)$ at these two endpoints and taking the maximum between them completes the proof. $\qquad\square$

Before proving Proposition 2.5, we need the following two results which asserts the analytical optimal value of maximizing a convex quadratic functions over a norm ball. These results can be found in the literature, the proof is included here for completeness.

**Lemma B.2** (Convex quadratic maximization over a norm ball). For any $\beta \in \mathbb{R}^m$, $\widehat{y} \in \mathbb{R}^m$ and $r \in \mathbb{R}_+$, the following assertions hold.

(i) Over a $\|\cdot\|_2$ ball, we have

$$
\sup \left\{ \|y - \beta\|_2^2 : \|y - \widehat{y}\|_2^2 \leq r^2 \right\} = (r + \|\widehat{y} - \beta\|_2)^2.
$$

(ii) Over a $\|\cdot\|_\infty$ ball, we have

$$
\sup \left\{ \|y - \beta\|_2^2 : \|y - \widehat{y}\|_\infty \leq r \right\} = \sum_{j\in[m]} \max \left\{ (\widehat{y}_j - \beta_j - r)^2, (\widehat{y}_j - \beta_j + r)^2 \right\},
$$

where $\beta_j$ and $\widehat{y}_j$ denote the $j$-th element of the vector $\beta$ and $\widehat{y}$, respectively.

91 *Proof of Lemma B.2.* We first prove Assertion (i). First, the optimal value is upper bounded by
92 $(r + \|\widehat{y} - \beta\|_2)^2$ because

$$\|y - \beta\|_2 \leq \|y - \widehat{y}\|_2 + \|\widehat{y} - \beta\|_2 \leq r + \|\widehat{y} - \beta\|_2$$

93 by triangle inequality. Yet, it is equal to that amount since that amount is attained when $y =$
94 $\widehat{y} + r(\widehat{y} - \beta)/\|\widehat{y} - \beta\|_2$.

95 Consider now Assertion (ii). Using a change of variables $z \leftarrow y - \beta$ and a change of parameters
96 $w \leftarrow \widehat{y} - \beta$, we find

$$\sup \left\{ \|y - \beta\|_2^2 : \|y - \widehat{y}\|_\infty \leq r \right\} = \max \left\{ \|z\|_2^2 : \|z - w\|_\infty \leq r \right\}, \tag{A.6}$$

97 where the maximization operators are justified by Weierstrass' maximum value theorem [1, Theo-
98 rem 2.43] because the feasible set is compact and the objective function is continuous. By extending
99 the norm constraint into the vector form, we have the equivalence

$$\max \left\{ \|z\|_2^2 : w - r\mathbb{1}_m \leq z \leq w + r\mathbb{1}_m \right\},$$

100 where the inequalities in the constraints are understood as element-wise inequalities, and $\mathbb{1}_m$ is an
101 $m$-dimensional vector of ones. This maximization problem is separable in the decision variables and
102 can be decomposed into $m$ independent univariate subproblems of the form

$$\max \left\{ z_j^2 : w_j - r \leq z_j \leq w_j + r \right\}$$

103 for each $j \in [m]$. It is easy to verify that the optimal value of each univariate subproblem is equal to

$$\max \left\{ (w_j - r)^2, (w_j + r)^2 \right\},$$

104 and summing up the optimal values over $j$ completes the proof. $\square$

105 We are now ready to prove Proposition 2.5.

106 *Proof of Proposition 2.5.* Following from equation (A.4a) in the proof of Theorem 2.3, we have

$$f(\beta) = \max \left\{ \frac{\sum_{i \in \mathcal{I}} v_i^\star(\beta)\alpha_i}{\sum_{i \in \mathcal{I}} \alpha_i} : \alpha \in [0, 1]^N, \ \alpha_i = 1 \ \forall i \in \mathcal{I}_1, \ \sum_{i \in \mathcal{I}} \alpha_i > 0 \right\}$$

107 By applying the Charnes-Cooper transformation [4] with

$$z_i = \frac{\alpha_i}{\sum_{i \in \mathcal{I}} \alpha_i}, \quad \text{and} \quad t = \frac{1}{\sum_{i \in \mathcal{I}} \alpha_i}$$

108 to reformulate this fractional linear problem, we have

$$f(\beta) = \begin{cases} \max & \sum_{i \in \mathcal{I}} v_i^\star(\beta) z_i \\ \text{s.t.} & \sum_{i \in \mathcal{I}} z_i = 1, \ t \geq 0 \\ & z_i - t = 0 & \forall i \in \mathcal{I}_1 \\ & 0 \leq z_i \leq t & \forall i \in \mathcal{I}_2. \end{cases}$$
$$= \begin{cases} \min & \lambda \\ \text{s.t.} & \lambda \in \mathbb{R}, \ u_i \in \mathbb{R} \ \forall i \in \mathcal{I}_1, \ u_i \in \mathbb{R}_+ \ \forall i \in \mathcal{I}_2 \\ & \lambda + u_i \geq v_i^\star(\beta) & \forall i \in \mathcal{I} \\ & \sum_{i \in \mathcal{I}} u_i \leq 0, \end{cases}$$

109 where the second equality follows from linear programming duality. Using the last minimization
110 reformulation of $f(\beta)$, problem (2) is now equivalent to

$$\min_\beta f(\beta) = \begin{cases} \min & \lambda \\ \text{s.t.} & \beta \in \mathbb{R}^m, \ \lambda \in \mathbb{R}, \ u_i \in \mathbb{R} \ \forall i \in \mathcal{I}_1, \ u_i \in \mathbb{R}_+ \ \forall i \in \mathcal{I}_2 \\ & \lambda + u_i \geq v_i^\star(\beta) & \forall i \in \mathcal{I} \\ & \sum_{i \in \mathcal{I}} u_i \leq 0, \end{cases}$$

111 When $\mathbb{D}_\mathcal{Y}$ is a 2-norm, each value $v_i^\star(\beta)$ calculated from (5) becomes

$$v_i^\star(\beta) = \sup \left\{ \|y - \beta\|_2^2 : \|y - \widehat{y}_i\|_2 \leq \rho - \mathbb{D}_\mathcal{X}(\widehat{x}_i^p, \widehat{x}_i) \right\} \qquad \forall i \in [N].$$

112    For any $i \in \mathcal{I}$, the value $v_i^\star(\beta)$ is finite and $v_i^\star(\beta)$ can be re-expressed by exploiting Lemma B.2(i) as

$$v_i^\star(\beta) = \left(\rho - \mathbb{D}_\mathcal{X}(\widehat{x}_i^p, \widehat{x}_i) + \|\widehat{y}_i - \beta\|_2\right)^2.$$

113    Problem (2) is now equivalent to

$$
\begin{aligned}
\min \quad & \lambda \\
\text{s.\,t.} \quad & \beta \in \mathbb{R}^m,\ \lambda \in \mathbb{R},\ u_i \in \mathbb{R}\ \forall i \in \mathcal{I}_1,\ u_i \in \mathbb{R}_+\ \forall i \in \mathcal{I}_2 \\
& \lambda + u_i \geq \left(\rho - \mathbb{D}_\mathcal{X}(\widehat{x}_i^p, \widehat{x}_i) + \|\widehat{y}_i - \beta\|_2\right)^2 \quad \forall i \in \mathcal{I} \\
& \sum_{i \in \mathcal{I}} u_i \leq 0.
\end{aligned}
\tag{A.7}
$$

114    To obtain a second-order cone program formulation, it now suffices to add the hypergraph formulation
115    $t_i \geq \|\widehat{y}_i - \beta\|_2$ with $t_i \geq 0$, and reformulate the quadratic constraint into a second-order cone
116    constraint using results from [2, Section 2]. This completes the proof for claim (i).

117    We now proceed to prove claim (ii). When $\mathbb{D}_\mathcal{Y}$ is the $\infty$-norm, each value $v_i^\star(\beta)$ becomes

$$v_i^\star(\beta) = \sup\left\{\|y - \beta\|_2^2 : \ \|y - \widehat{y}_i\|_\infty \leq \rho - \mathbb{D}_\mathcal{X}(\widehat{x}_i^p, \widehat{x}_i)\right\} \qquad \forall i \in [N].$$

118    For any $i \in \mathcal{I}$, the value $v_i^\star(\beta)$ is finite and $v_i^\star(\beta)$ can be re-expressed using Lemma B.2(ii) as

$$v_i^\star(\beta) = \sum_{j \in [m]} \max\left\{(\widehat{y}_{ij} - \beta_j - \rho + \mathbb{D}_\mathcal{X}(\widehat{x}_i^p, \widehat{x}_i))^2, (\widehat{y}_{ij} - \beta_j + \rho - \mathbb{D}_\mathcal{X}(\widehat{x}_i^p, \widehat{x}_i))^2\right\}.$$

119    By adding auxiliary variables $T_{ij}$ with the constraints

$$(\widehat{y}_{ij} - \beta_j - \rho + \mathbb{D}_\mathcal{X}(\widehat{x}_i^p, \widehat{x}_i))^2 \leq T_{ij}^2, \quad \text{and} \quad (\widehat{y}_{ij} - \beta_j + \rho - \mathbb{D}_\mathcal{X}(\widehat{x}_i^p, \widehat{x}_i))^2 \leq T_{ij}^2,$$

120    problem (2) is now equivalent to

$$
\begin{aligned}
\min \quad & \lambda \\
\text{s.\,t.} \quad & \beta \in \mathbb{R}^m,\ \lambda \in \mathbb{R},\ T \in \mathbb{R}_+^{|\mathcal{I}| \times m},\ u_i \in \mathbb{R}\ \forall i \in \mathcal{I}_1,\ u_i \in \mathbb{R}_+\ \forall i \in \mathcal{I}_2 \\
& \sum_{i \in \mathcal{I}} u_i \leq 0 \\
& \lambda + u_i \geq \sum_{j \in [m]} T_{ij}^2 \quad \forall i \in \mathcal{I} \\
& (\widehat{y}_{ij} - \beta_j - \rho + \mathbb{D}_\mathcal{X}(\widehat{x}_i^p, \widehat{x}_i))^2 \leq T_{ij}^2 && \forall (i,j) \in \mathcal{I} \times [m] \\
& (\widehat{y}_{ij} - \beta_j + \rho - \mathbb{D}_\mathcal{X}(\widehat{x}_i^p, \widehat{x}_i))^2 \leq T_{ij}^2 && \forall (i,j) \in \mathcal{I} \times [m].
\end{aligned}
$$

121    The last two constraints can be re-expressed as linear constraints of the form

$$
\begin{aligned}
-T_{ij} \leq \widehat{y}_{ij} - \beta_j - \rho + \mathbb{D}_\mathcal{X}(\widehat{x}_i^p, \widehat{x}_i) \leq T_{ij} \quad \forall (i,j) \in \mathcal{I} \times [m] \\
-T_{ij} \leq \widehat{y}_{ij} - \beta_j + \rho - \mathbb{D}_\mathcal{X}(\widehat{x}_i^p, \widehat{x}_i) \leq T_{ij} \quad \forall (i,j) \in \mathcal{I} \times [m].
\end{aligned}
$$

122    Formulating the quadratic constraint $\lambda + u_i \geq \sum_{j \in [m]} T_{ij}^2$ using [2, Section 2] completes the
123    proof. $\qquad\qquad\square$

124    *Proof of Proposition 2.6.* For the purpose of this proof, define the following sets

$$\mathcal{Y}_i \triangleq \{y_i \in \mathcal{Y} : \mathbb{D}_\mathcal{Y}(y_i, \widehat{y}_i) \leq \rho - \mathbb{D}_\mathcal{X}(\widehat{x}_i^p, \widehat{x}_i)\} \qquad \forall i \in \mathcal{I}.$$

125    Because $\mathbb{D}_\mathcal{Y}$ is coercive and continuous, each set $\mathcal{Y}_i$ is compact. Because the loss function is
126    continuous, there thus exists $y_i^\star$ satisfying $y_i^\star \in \mathcal{Y}_i$ and $\ell(y_i^\star, \beta) = v_i^\star(\beta)$ for any $i \in \mathcal{I}$. Following
127    from Equation (A.4a) in the proof of Theorem 2.3, we have

$$
\begin{aligned}
f(\beta) &= \max\left\{\frac{\sum_{i \in \mathcal{I}} v_i^\star(\beta)\alpha_i}{\sum_{i \in \mathcal{I}} \alpha_i} : \alpha \in [0,1]^N,\ \alpha_i = 1\ \forall i \in \mathcal{I}_1,\ \sum_{i \in \mathcal{I}} \alpha_i > 0\right\} \\
&= \max\left\{\frac{\sum_{i \in \mathcal{I}} \ell(y_i, \beta)\alpha_i}{\sum_{i \in \mathcal{I}} \alpha_i} : \alpha \in [0,1]^N,\ \alpha_i = 1\ \forall i \in \mathcal{I}_1,\ \sum_{i \in \mathcal{I}} \alpha_i > 0,\ y_i \in \mathcal{Y}_i\ \forall i \in \mathcal{I}\right\}.
\end{aligned}
$$

128    If $\mathcal{I}_1 = \emptyset$, then we have

$$f(\beta) = \ell(y_{i^\star}, \beta) \quad \forall i^\star \in \arg\max_{i \in \mathcal{I}_2} v_i^\star(\beta),$$

129    and a subgradient of $f$ is $\partial f(\beta) = \partial_\beta \ell(y_{i^\star}, \beta)$ for any $i^\star \in \arg\max_{i \in \mathcal{I}_2} v_i^\star(\beta)$. By incorporating
130    the optimal value of $\alpha$ in the statement of Theorem 2.3, we have $\partial f(\beta) = \alpha_i \partial_\beta \ell(y_i^\star, \beta)$.

131    If $\mathcal{I}_1 \neq \emptyset$, then we have

$$f(\beta) = \max \left\{ \frac{\sum_{i \in \mathcal{I}_1} v_i^\star(\beta) + \sum_{i \in \mathcal{I}_2} v_i^\star(\beta)\alpha_i}{|\mathcal{I}_1| + \sum_{i \in \mathcal{I}_2} \alpha_i} : \alpha \in [0,1]^N, \ \alpha_i = 1 \ \forall i \in \mathcal{I}_1 \right\}$$

$$= \max \left\{ \frac{\sum_{i \in \mathcal{I}_1} \ell(y_i, \beta) + \sum_{i \in \mathcal{I}_2} \ell(y_i, \beta)\alpha_i}{|\mathcal{I}_1| + \sum_{i \in \mathcal{I}_2} \alpha_i} : \alpha \in [0,1]^N, \ \alpha_i = 1 \ \forall i \in \mathcal{I}_1, \ y_i \in \mathcal{Y}_i \ \forall i \in \mathcal{I} \right\}$$

132    Notice that the function

$$\beta \mapsto \frac{\sum_{i \in \mathcal{I}_1} \ell(y_i, \beta) + \sum_{i \in \mathcal{I}_2} \ell(y_i, \beta)\alpha_i}{|\mathcal{I}_1| + \sum_{i \in \mathcal{I}_2} \alpha_i}$$

133    is convex for any feasible value of $(\alpha, y)$ in the above optimization problem. Moreover, by Ty-
134    chonoff's theorem [1, Theorem 2.61], the feasible set of the above optimization problem is a compact
135    set in the product topology. One can now apply [3, Proposition A.22] to conclude that a subgradient
136    of $f$ in this case is

$$\partial f(\beta) = \frac{\sum_{i \in \mathcal{I}_1} \partial_\beta \ell(y_i, \beta) + \sum_{i \in \mathcal{I}_2} \partial_\beta \ell(y_i, \beta)\alpha_i}{|\mathcal{I}_1| + \sum_{i \in \mathcal{I}_2} \alpha_i}.$$

137    Combining the two cases, we have the postulated result.      $\square$

## 138   B.2   Proofs of Section 3

139    *Proof of Proposition 3.1.* Under the conditions of the proposition, we have $\mathbb{P}(X \in \mathcal{N}_\gamma(x_0)) > 0$
140    because $\mathbb{P}$ admits a density, and that $\mathcal{N}_\gamma(x_0) \cap \mathcal{X}$ is a set with non-empty interior for any $\gamma > 0$. The
141    proof now follows trivially from [5, Theorem 1.1]. Indeed, under the conditions of the proposition,
142    with probability of at least $1 - O(N^{-c})$, we have $\mathbb{P} \in \mathbb{B}_\rho^\infty$, and hence the bound follows.      $\square$

143    *Proof of Example 3.2.* For the purpose of this proof, we let $\mathbb{P}^\infty = \mathbb{P} \otimes \mathbb{P} \otimes \cdots$ be the joint distribution
144    of $(\widehat{x}_1, \widehat{y}_1), (\widehat{x}_2, \widehat{y}_2), \cdots$. The selection of parameter $\gamma = 0$ implies that $\mathcal{I} = \mathcal{I}_2$, and for any fixed
145    $\rho > 0$ we have $\mathbb{P}^\infty (\lim_{N \to \infty} |\mathcal{I}| = +\infty) = 1$ by Borel-Cantelli lemma. In this example, the DRO
146    problem is feasible if $\mathcal{I}$ is nonempty, and we have an explicit optimal solution

$$\beta_N^\star = \frac{1}{2} \min_{i \in \mathcal{I}} \{\widehat{y}_i - \rho + \mathbb{D}_{\mathcal{X}}(\widehat{x}_i, x_0)\} + \frac{1}{2} \max_{i \in \mathcal{I}} \{\widehat{y}_i + \rho - \mathbb{D}_{\mathcal{X}}(\widehat{x}_i, x_0)\}$$

147    Notice that with probability 1 we have

$$\min_{i \in \mathcal{I}} \{\widehat{y}_i - \rho + \mathbb{D}_{\mathcal{X}}(\widehat{x}_i, x_0)\} \geq -\rho \text{ and } \max_{i \in \mathcal{I}} \{\widehat{y}_i + \rho - \mathbb{D}_{\mathcal{X}}(\widehat{x}_i, x_0)\} \geq \max_{i \in \mathcal{I}} \{\widehat{y}_i\}.$$

148    Consequently we have $\beta_N^\star \geq \frac{1}{2} \max_{i \in \mathcal{I}} \{\widehat{y}_i\} - \frac{1}{2}\rho$. For all $y > 0$, we have

$$\mathbb{P}^\infty \left( \lim_{N \to \infty} \beta_N^\star > y \right) \geq \mathbb{P}^\infty \left( \lim_{N \to \infty} \max_{i \in \mathcal{I}} \{\widehat{y}_i\} > 2y + \rho \right) = \lim_{N \to \infty} \mathbb{P}^\infty \left( \max_{i \in \mathcal{I}} \{\widehat{y}_i\} > 2y + \rho \right)$$

$$= \lim_{N \to \infty} 1 - \mathbb{P}(Y \leq 2y + \rho)^{|\mathcal{I}|} = 1.$$

149    Let $y$ tend to infinity concludes the proof.      $\square$

150    Before proving Proposition 3.4, we first present the following minimax result.

151    **Lemma B.3** (Minimax result). Suppose that $\ell(y, \cdot)$ is convex and coercive for any $y \in \mathcal{Y}$, and that
152    $\mathbb{D}_{\mathcal{Y}}(\cdot, \widehat{y})$ is convex and coercive for any $\widehat{y}$. For any $\rho \geq \min_{i \in [N]} \kappa_{i,\gamma}$, we have

$$\min_{\beta \in \mathbb{R}^m} \sup_{\mathbb{Q} \in \mathbb{B}_\rho^\infty, \mathbb{Q}(X \in \mathcal{N}_\gamma(x_0)) > 0} \mathbb{E}_{\mathbb{Q}}\big[\ell(Y, \beta) | X \in \mathcal{N}_\gamma(x_0)\big]$$

$$= \sup_{\mathbb{Q} \in \mathbb{B}_\rho^\infty, \mathbb{Q}(X \in \mathcal{N}_\gamma(x_0)) > 0} \min_{\beta \in \mathbb{R}^m} \mathbb{E}_{\mathbb{Q}}\big[\ell(Y, \beta) | X \in \mathcal{N}_\gamma(x_0)\big].$$

153    To facilitate the proof of Lemma B.3, we define the following conditional ambiguity set induced by
154    $\mathbb{B}_\rho^\infty$ as

$$\mathcal{B}_{x_0, \gamma}(\mathbb{B}_\rho^\infty) \triangleq \left\{ \mu_0 \in \mathcal{M}(\mathcal{Y}) : \begin{array}{l} \exists \mathbb{Q} \in \mathbb{B}_\rho^\infty, \ \mathbb{Q}(\mathcal{N}_\gamma(x_0) \times \mathcal{Y}) > 0 \\ \mathbb{Q}(\mathcal{N}_\gamma(x_0) \times A) = \mu_0(A) \, \mathbb{Q}(\mathcal{N}_\gamma(x_0) \times \mathcal{Y}) \ \forall A \subseteq \mathcal{Y} \text{ measurable} \end{array} \right\},$$

(A.8)

155 where the last constraint defining the set $\mathcal{B}_{x_0,\gamma}(\mathbb{B}_\rho^\infty)$ is from the dis-integration of the joint measure
156 into a marginal distribution and the corresponding conditional distributions [8, Theorem 9.2.2].

157 The proof of Lemma B.3 relies on the following two results which assert the convexity of the joint
158 ambiguity set $\mathbb{B}_\rho^\infty$ and its induced conditional ambiguity set $\mathcal{B}_{x_0,\gamma}(\mathbb{B}_\rho^\infty)$.

159 **Lemma B.4** (Convexity of $\mathbb{B}_\rho^\infty$)**.** The ambiguity set $\mathbb{B}_\rho^\infty$ is convex.

160 *Proof of Lemma B.4.* Because the nominal probability measure is an empirical measure, the ambigu-
161 ity set $\mathbb{B}_\rho^\infty$ can be represented as

$$\mathbb{B}_\rho^\infty = \left\{ \mathbb{Q} \in \mathcal{M}(\mathcal{X} \times \mathcal{Y}) : \begin{array}{l} \exists \pi_i \in \mathcal{M}(\mathcal{X} \times \mathcal{Y}) \; \forall i \in [N] \text{ such that :} \\ \mathbb{Q} = N^{-1}\sum_{i\in[N]}\pi_i, \; \sum_{i\in[N]}\pi_i(\mathcal{N}_\gamma(x_0) \times \mathcal{Y}) > 0 \\ \mathbb{D}_{\mathcal{X}}(x,\widehat{x}_i) + \mathbb{D}_{\mathcal{Y}}(y,\widehat{y}_i) \le \rho \quad \forall(x,y) \in \mathrm{supp}(\pi_i) \quad \forall i \in [N] \end{array} \right\}.$$

162 Pick any arbitrary $\mathbb{Q}^0$ and $\mathbb{Q}^1$ from $\mathbb{B}_\rho^\infty$. Associated with $\mathbb{Q}^j$, $j \in \{0,1\}$ is a collection of probability
163 measures $\{\pi_i^j\} \in \mathcal{M}(\mathcal{X} \times \mathcal{Y})^N$ satisfying

$$\left\{ \begin{array}{l} \mathbb{Q}^j = N^{-1}\sum_{i\in[N]}\pi_i^j, \; \sum_{i\in[N]}\pi_i^j(\mathcal{N}_\gamma(x_0) \times \mathcal{Y}) > 0 \\ \mathbb{D}_{\mathcal{X}}(x,\widehat{x}_i) + \mathbb{D}_{\mathcal{Y}}(y,\widehat{y}_i) \le \rho \quad \forall(x,y) \in \mathrm{supp}(\pi_i^j) \quad \forall i \in [N]. \end{array} \right.$$

164 Consider any convex combination $\mathbb{Q}^\lambda = \lambda\mathbb{Q}^1 + (1-\lambda)\mathbb{Q}^0$ for $\lambda \in (0,1)$. It is easy to verify that
165 the joint measure $\pi_i^\lambda = \lambda\pi_i^1 + (1-\lambda)\pi_i^0$ for any $i \in [N]$ satisfies

$$\left\{ \begin{array}{l} \mathbb{Q}^\lambda = N^{-1}\sum_{i\in[N]}\pi_i^\lambda, \; \sum_{i\in[N]}\pi_i^\lambda(\mathcal{N}_\gamma(x_0) \times \mathcal{Y}) > 0 \\ \mathbb{D}_{\mathcal{X}}(x,\widehat{x}_i) + \mathbb{D}_{\mathcal{Y}}(y,\widehat{y}_i) \le \rho \quad \forall(x,y) \in \mathrm{supp}(\pi_i^\lambda) \quad \forall i \in [N], \end{array} \right.$$

166 where the last constraint is satisfied by noticing that $\mathrm{supp}(\pi_i^\lambda) = \mathrm{supp}(\pi_i^0) \cup \mathrm{supp}(\pi_i^1)$. This
167 observation implies that $\mathbb{Q}^\lambda \in \mathbb{B}_\rho^\infty$. $\square$

168 **Lemma B.5** (Convexity of $\mathcal{B}_{x_0,\gamma}(\mathbb{B}_\rho^\infty)$)**.** The conditional ambiguity set $\mathcal{B}_{x_0,\gamma}(\mathbb{B}_\rho^\infty)$ is convex.

169 *Proof of Lemma B.5.* Let $\mu_0^0, \mu_0^1 \in \mathcal{B}_{x_0,\gamma}(\mathbb{B}_\rho^\infty)$ be two arbitrary probability measures. Associated
170 with each $\mu_0^j$, $j \in \{0,1\}$, is a corresponding joint measure $\mathbb{Q}^j \in \mathcal{M}(\mathcal{X} \times \mathcal{Y})$ such that

$$\mathbb{Q}^j(\mathcal{N}_\gamma(x_0) \times \mathcal{Y}) > 0 \quad \text{and} \quad \frac{\mathbb{Q}^j(\mathcal{N}_\gamma(x_0) \times A)}{\mathbb{Q}^j(\mathcal{N}_\gamma(x_0) \times \mathcal{Y})} = \mu_0^j(A).$$

171 Select any $\lambda \in (0,1)$. We proceed to show that $\mu_0^\lambda = \lambda\mu_0^1 + (1-\lambda)\mu_0^0 \in \mathcal{B}_{x_0,\gamma}(\mathbb{B}_\rho^\infty)$. Indeed,
172 consider the joint measure
$$\mathbb{Q}^\lambda = \theta\mathbb{Q}^1 + (1-\theta)\mathbb{Q}^0$$

173 with $\theta$ being defined as

$$\theta = \frac{\lambda\mathbb{Q}^0(\mathcal{N}_\gamma(x_0) \times \mathcal{Y})}{\lambda\mathbb{Q}^0(\mathcal{N}_\gamma(x_0) \times \mathcal{Y}) + (1-\lambda)\mathbb{Q}^1(\mathcal{N}_\gamma(x_0) \times \mathcal{Y})} \in [0,1].$$

174 By definition, we have $\mathbb{Q}^\lambda(\mathcal{N}_\gamma(x_0) \times \mathcal{Y}) > 0$, and by convexity of $\mathbb{B}_\rho^\infty$ from Lemma B.4, we have
175 $\mathbb{Q}^\lambda \in \mathbb{B}_\rho^\infty$. Moreover, we have

$$\begin{aligned} \frac{\mathbb{Q}^\lambda(\mathcal{N}_\gamma(x_0) \times A)}{\mathbb{Q}^\lambda(\mathcal{N}_\gamma(x_0) \times \mathcal{Y})} &= \frac{\theta\mathbb{Q}^1(\mathcal{N}_\gamma(x_0) \times A) + (1-\theta)\mathbb{Q}^0(\mathcal{N}_\gamma(x_0) \times A)}{\theta\mathbb{Q}^1(\mathcal{N}_\gamma(x_0) \times \mathcal{Y}) + (1-\theta)\mathbb{Q}^0(\mathcal{N}_\gamma(x_0) \times \mathcal{Y})} \\ &= \frac{\lambda\mathbb{Q}^0(\mathcal{N}_\gamma(x_0) \times \mathcal{Y})\mathbb{Q}^1(\mathcal{N}_\gamma(x_0) \times A) + (1-\lambda)\mathbb{Q}^1(\mathcal{N}_\gamma(x_0) \times \mathcal{Y})\mathbb{Q}^0(\mathcal{N}_\gamma(x_0) \times A)}{\mathbb{Q}^0(\mathcal{N}_\gamma(x_0) \times \mathcal{Y})\mathbb{Q}^1(\mathcal{N}_\gamma(x_0) \times \mathcal{Y})} \\ &= \frac{\lambda\mathbb{Q}^1(\mathcal{N}_\gamma(x_0) \times A)}{\mathbb{Q}^1(\mathcal{N}_\gamma(x_0) \times \mathcal{Y})} + \frac{(1-\lambda)\mathbb{Q}^0(\mathcal{N}_\gamma(x_0) \times A)}{\mathbb{Q}^0(\mathcal{N}_\gamma(x_0) \times \mathcal{Y})} \\ &= \lambda\mu_0^1(A) + (1-\lambda)\mu_0^0(A), \end{aligned}$$

176 where the second equality follows from the definition of $\theta$. This implies that $\mu_0^\lambda \in \mathcal{B}_{x_0,\gamma}(\mathbb{B}_\rho^\infty)$, and
177 further implies the convexity of $\mathcal{B}_{x_0,\gamma}(\mathbb{B}_\rho^\infty)$. $\square$

178  We are now ready to prove Lemma B.3.

179  *Proof of Lemma B.3.* By the definition of the conditional ambiguity set $\mathcal{B}_{x_0,\gamma}(\mathbb{B}_\rho^\infty)$, it suffices to
180  prove the equivalence

$$\min_{\beta\in\mathbb{R}^m}\ \sup_{\mu_0\in\mathcal{B}_{x_0,\gamma}(\mathbb{B}_\rho^\infty)}\ \mathbb{E}_{\mu_0}[\ell(Y,\beta)] = \sup_{\mu_0\in\mathcal{B}_{x_0,\gamma}(\mathbb{B}_\rho^\infty)}\ \min_{\beta\in\mathbb{R}^m}\ \mathbb{E}_{\mu_0}[\ell(Y,\beta)].$$

181  First, consider the mapping $\beta\mapsto\sup_{\mu_0\in\mathcal{B}_{x_0,\gamma}(\mathbb{B}_\rho^\infty)}\mathbb{E}_{\mu_0}[\ell(Y,\beta)]$. The properties of $\ell$ implies that
182  this mapping is lower semi-continuous and coercive. As a consequence, without loss of optimality,
183  we can restrict the feasible set $\beta$ to some convex, compact ball $\mathcal{S}\triangleq\{\beta:\|\beta\|_2\le R\}$ for some radius
184  $R\in\mathbb{R}_{++}$ sufficiently big.

185  We now consider the mapping $\mu_0\mapsto\mathbb{E}_{\mu_0}[\ell(Y,\beta)]$ parametrized by $\beta$. For any $\beta$, it is a linear function
186  of $\mu_0$, and hence it is concave. It is also weakly continuous. To see this, notice that when $\mathbb{D}(\cdot,\widehat{y})$ is
187  coercive, the set

$$\mathcal{A}\triangleq\bigcup_{i\in[N]}\{y:\mathbb{D}_\mathcal{Y}(y,\widehat{y}_i)\le\rho\},$$

188  being a finite union of bounded sets, is bounded. Pick any $\mathbb{Q}\in\mathbb{B}_\rho^\infty$, by the definition of the type-$\infty$
189  Wasserstein distance, we have $\mathbb{Q}(\mathcal{A})=1$. Consider the conditional measure $\mu_0^{\mathbb{Q}}$ induced by $\mathbb{Q}$, then
190  we have

$$\mu_0^{\mathbb{Q}}(\mathcal{A}\cap\mathcal{Y}) = \frac{\mathbb{Q}(\mathcal{N}_\gamma(x_0)\times(\mathcal{A}\cap\mathcal{Y}))}{\mathbb{Q}(\mathcal{N}_\gamma(x_0)\times\mathcal{Y})} \ge \frac{\mathbb{Q}(\mathcal{N}_\gamma(x_0)\times(\mathcal{A}\cap\mathcal{Y}))}{\mathbb{Q}(\mathcal{N}_\gamma(x_0)\times(\mathcal{A}\cap\mathcal{Y}))} = 1,$$

191  which implies that $\mu_0^{\mathbb{Q}}$ has a bounded support. This implies that $\mathcal{B}_{x_0,\gamma}(\mathbb{B}_\rho^\infty)\subseteq\mathcal{M}(\mathcal{A})$, where $\mathcal{M}(\mathcal{A})$
192  is the set of all probability measures supported on a bounded set $\mathcal{A}$. Because $\ell(\cdot,\beta)$ is continuous,
193  there exists a bound $U\in\mathbb{R}_{++}$ such that $|\ell(y,\beta)|\le U$ for every $y\in\mathcal{A}$. Define now the function
194  $\ell_U(\cdot,\beta)=\max\{-U,\min\{\ell(\cdot,\beta),U\}\}$, which is continuous and bounded. Consider any sequence
195  of conditional measures $\{\mu_0^k\}\in\mathcal{M}(\mathcal{A})$ that weakly converges to $\mu_0^\infty$, we have

$$\lim_{k\uparrow\infty}\mathbb{E}_{\mu_0^k}[\ell(Y,\beta)] = \lim_{k\uparrow\infty}\mathbb{E}_{\mu_0^k}[\ell_U(Y,\beta)] = \mathbb{E}_{\mu_0^\infty}[\ell_U(Y,\beta)] = \mathbb{E}_{\mu_0^\infty}[\ell(Y,\beta)],$$

196  which implies that the function $\mu_0\mapsto\mathbb{E}_{\mu_0}[\ell(Y,\beta)]$ is weakly continuous over $\mathcal{M}(\mathcal{A})$.

197  This line of argument suggests that

$$\min_\beta\ \sup_{\mu_0\in\mathcal{B}_{x_0,\gamma}(\mathbb{B}_\rho^\infty)}\ \mathbb{E}_{\mu_0}[\ell(Y,\beta)] = \min_{\beta:\|\beta\|_2\le R}\ \sup_{\mu_0\in\mathcal{B}_{x_0,\gamma}(\mathbb{B}_\rho^\infty)}\ \mathbb{E}_{\mu_0}[\ell(Y,\beta)]$$

$$= \sup_{\mu_0\in\mathcal{B}_{x_0,\gamma}(\mathbb{B}_\rho^\infty)}\ \min_{\beta:\|\beta\|_2\le R}\ \mathbb{E}_{\mu_0}[\ell(Y,\beta)] \qquad\text{(A.9a)}$$

$$= \sup_{\mu_0\in\mathcal{B}_{x_0,\gamma}(\mathbb{B}_\rho^\infty)}\ \min_\beta\ \mathbb{E}_{\mu_0}[\ell(Y,\beta)], \qquad\text{(A.9b)}$$

198  where equality (A.9b) follows from the coercivity of the loss function, thus the constraint on $\beta$ can be
199  dropped for $R$ sufficiently big. Equality (A.9a) holds by Sion's minimax theorem [7]. This finishes
200  the proof.  $\qquad\square$

201  *Proof of Proposition 3.4.* Because the loss function is coercive and convex in $\beta$, we have

$$\min_{\beta\in\mathbb{R}}\ \sup_{\mu_0\in\mathcal{B}_{x_0,\gamma}(\mathbb{B}_\rho^\infty)}\ \mathbb{E}_{\mu_0}[(Y-\beta)^2] = \sup_{\mu_0\in\mathcal{B}_{x_0,\gamma}(\mathbb{B}_\rho^\infty)}\ \min_{\beta\in\mathbb{R}}\ \mathbb{E}_{\mu_0}[(Y-\beta)^2]$$

$$= \sup_{\mu_0\in\mathcal{B}_{x_0,\gamma}(\mathbb{B}_\rho^\infty)}\ \mathbb{E}_{\mu_0}[(Y-\mathbb{E}_{\mu_0}[Y])^2]$$

$$= \text{Variance}_{\mu_0^\star}(Y),$$

202  where the first equality follows from Lemma B.3, the second equality follows from the fact that for
203  any $\mu_0\in\mathcal{B}_{x_0,\gamma}(\mathbb{B}_\rho^\infty)$, the estimate $\beta^\star(\mu_0)=\mathbb{E}_{\mu_0}[Y]$ minimizes the objective $\mathbb{E}_{\mu_0}[(Y-\beta)^2]$. The
204  last equality follows from the definition of $\mu_0^\star$.

Let $\beta^\star$ be the optimal estimate that solves (2), we now have

$$\text{Variance}_{\mu_0^\star}(Y) = \sup_{\mu_0 \in \mathcal{B}_{x_0,\gamma}(\mathbb{B}_\rho^\infty)} \mathbb{E}_{\mu_0}[(Y - \beta^\star)^2]$$

$$\geq \mathbb{E}_{\mu_0^\star}[(Y - \beta^\star)^2] = \text{Variance}_{\mu_0^\star}(Y) + (\beta^\star - \mathbb{E}_{\mu_0^\star}[Y])^2,$$

where the last equality follows from the bias-variance decomposition. This implies that $\beta^\star = \mathbb{E}_{\mu_0^\star}[Y]$ and completes the proof. $\square$

## C  Golden-section Search for Univariate Conditional Estimate

We elaborate here on the procedure of applying a golden-section search to solve a one-dimensional local conditional estimation with a convex loss function $\ell$. We suppose that $\mathcal{Y} = [a, b]$ for some finite values $-\infty < a < b < \infty$, that $\ell(y, \cdot)$ is convex for every $y$ and that we have access to an oracle that solves (5). Given any $\beta$, the worst-case conditional expected loss $f(\beta)$ can be computed using Theorem 2.3. Algorithm 1 can be used to find the optimal conditional estimate $\beta^\star$ to any arbitrary precision.

---

**Algorithm 1** Golden-section Search Algorithm

---

**Input:** Range $[a, b] \in \mathbb{R}$, tolerance $\epsilon \in \mathbb{R}_{++}$
**Initialization:** Set $r \leftarrow 0.618$, $\beta_1 \leftarrow a$, $\beta_4 \leftarrow b$
**while** $|\beta_4 - \beta_1| > \epsilon$ **do**
    Set $\beta_2 \leftarrow r\beta_1 + (1 - r)\beta_4$, $\beta_3 \leftarrow (1 - r)\beta_1 + r\beta_4$
    **if** $f(\beta_2) \leq f(\beta_3)$ **then** Set $\beta_4 \leftarrow \beta_3$ **else** Set $\beta_1 \leftarrow \beta_2$ **endif**
**end while**
Set $\beta^\star \leftarrow (\beta_1 + \beta_4)/2$
**Output:** $\beta^\star$

---

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

Figure A.4: Comparison of estimations from N-W and DRCME on entropic regularized Wasserstein barycenters of pairs of images from the training set. Estimations are presented above each image in the format "(N-W, DRCME )".

Figure A.5: Comparison of estimations from BertEtAl and DRCME on entropic regularized Wasserstein barycenters of pairs of images from the training set. Estimations are presented above each image in the format "(BertEtAl , DRCME )".