[Reviews · NeurIPS 2020]

Review 1

Summary and Contributions: This paper considers the problem of non-parametric conditional estimation such as nearest neighbor regression in the presence of adversarial examples. Following some existing literature, it formulates the problem as distributional robust optimization and derives a solution. For conditional mean estimation the solution reduces to a convex program. Some analysis of rate of convergence is then provided under the condition that the conditional distribution is smooth.

Strengths: + The problem is timely. While there has been quite a bit of work on adversarially robust nearest neighbors and other non-parametric (see below for missing literature review), the problem is far from solved. This paper presents a reasonable solution. + The proposed method appears to be technically correct. + While DRO and perturbations wrt infinity Wasserstein balls have been used in the context of adversarial examples and classification with neural networks before, it is a new approach for non-parametric estimation. So the formulation and the results are a novel alternative way to look at the problem.

Weaknesses: - The approach seems to be somewhat non-transparent -- for example, for nearest neighbors, it is a little unclear what exactly the solution is doing, and why averaging over such a large ball is needed. I would recommend adding some discussion on this. - While the paper's approach is different, it is missing comparisons to recent theoretical and empirical work on robustness of nearest neighbors and other non-parametric methods. See below for more details.

Correctness: Methodology appears to be correct although I did not verify the proofs.

Clarity: Mostly well-written -- although the approach could be made more transparent.

Relation to Prior Work: The paper is missing references to and comparisons with important related works on adversarial examples for nearest neighbors and other non-parametric methods. For example, [1] provides a direct convergence rate for robustness of nearest neighbors to adversarial examples; it would be good to discuss how the bounds in this paper are different. Similarly, the convex program proposed by this paper feels similar to [5] as well as [2]; it would be good to discuss the relationship between these works. [1] Analyzing the robustness of nearest neighbors to adversarial examples, ICML 2018. [2] On the Geometry of Adversarial Examples, https://arxiv.org/abs/1811.00525 [3] Robustness for non-parametric methods: a defense and an attack, AISTATS 2020. [4] When are non-parametric methods robust?, ICML 2020. [5] Evaluating the Robustness of Nearest Neighbor Classifiers: A Primal-Dual Perspective, https://arxiv.org/abs/1906.03972

Reproducibility: Yes

Additional Feedback: Overall this is a solid paper, and a good fit for the conference. The paper could be significantly strengthened by making the approach more transparent and citing, and discussing directly related previous work. -- The authors have addressed my questions, and my opinion (accept) remains unchanged. I urge the authors to add these discussions and the context of prior work to the final version of the paper.


Review 2

Summary and Contributions: Upon reading the response and the other reviews, I raised my score. I am still not entirely convinced about the strength of empirical evaluation. Even though MNIST is a benchmark dataset, the proposed task does not seem like a natural setting for non-parametric conditional estimation. ------------- The paper introduces a new estimator for conditional statistics (such as the conditional mean). The estimator uses distributionally robust optimization to ensure robustness of the estimator. The authors demonstrate that under certain assumptions the estimator is efficiently computable, and they also provide generalization and consistency guarantees.

Strengths: The analysis of the computational efficiency of the proposed estimator is interesting and novel.

Weaknesses: While I agree that many tasks in machine learning can be cast as conditional estimation problems, I would have liked to see a concrete setting to which the estimator is particularly well suited, and where it can be shown (either theoretically or empirically) to perform competitively with existing methods. I found the empirical evaluation to be very limited, because: 1) both the studied tasks (mean estimation with synthetic data, and digit estimation on MNIST) are of minimal practical interest, and 2) the numerical results are not very convincing. The theoretical contributions of the paper involve demonstrating that the proposed estimator is under certain settings tractable and consistent, but fail to illustrate its strengths.

Correctness: Yes.

Clarity: Yes.

Relation to Prior Work: Much of the discussion of related work feels out of place and overly broad.

Reproducibility: Yes

Additional Feedback:


Review 3

Summary and Contributions: This paper discusses the problem of robust non-parametric conditional estimation for mean, quantiles, and alike. They formulate the problem using infinity Wasserstein ball around the empricial distribution and under some conditions prove consistency, upper bounds and computational complexity of the problem. The theoretical arguments are interesting on their own, and they also perform experiments on MNIST and synthetic examples and compare their approach with other estimators.

Strengths: The theory developed is nice, especially I found Proposition 2.2, Theorem 2.3, and Proposition 3.4 very interesting. It is a novel application of simple methods in optimization applied to a statistical problem. The problem is at the very heart of machine learning and statistics and can have huge impacts on the community. The numerical experiment was very insightful and showed how other estimators behave in terms of robustness.

Weaknesses: Some theory developed is incomplete and needs further toughts. The result of Proposition 3.1 just provides an upperbound on the expected value of the estimate, while no lowerbounds are provided. Moreover, the result of Example 3.3 is not very interesting, as it is under very tight conditions (linearity of the conditional expectation with respect to x), and a common use-case is not provided. The experimental results on MNIST is not very pleasing and does not show the superiority against NW estimate.

Correctness: The results, as far as I understood and skimmed through the appendix, seems correct, however, I did not verfied the proofs line by line.

Clarity: Yes, it is a well written paper.

Relation to Prior Work: Yes, it is clear what others miss in terms of robustness or their assumptions.

Reproducibility: Yes

Additional Feedback: ===== EDIT ===== after reading the authors' rebuttal, I am convinced that the score I gave is fair.

[Author Response · NeurIPS 2020]

We would like to thank all referees for their appreciation of our results and the useful feedback. Below is our reply.

**Reviewer 1:** There are (at least) two reasons to justify the averaging over a ball of radius $\gamma > 0$ around $x_0$. First, Example 3.2 indicates that when $\gamma = 0$, the estimator may be inconsistent. This is equivalent to 1-nearest neighbor is not consistent in general. Second, Example 3.3 shows that we can recover the $k$-nearest neighbors by choosing $\gamma$ appropriately. Averaging over a ball around $x_0$ thus eliminates an undesirable statistical property (inconsistency) and gives us the flexibility to recover $k$-nearest neighbors for general $k$.

To improve the transparency of our estimator, we will provide in the revision a description of the worst-case distribution. Just as an adversarial example provides a description on how to perturb a data point, the worst-case distribution provides full information on how to perturb the empirical distribution from the adversary's viewpoint. For our estimator, constructing the worst-case distribution is intriguing and intuitive: it involves sorting the values $v^\star$ defined in equation (5) and then performing a greedy assignment. The construction of this worst-case distribution is done as part of the proof of Theorem 2.3. We agree that this information should be made more explicit to the readers. We will include the worst-case distribution and elaborate more details in the revised version.

Thank you for pointing out the relevant literature. To our understanding, existing robustification of nearest neighbors (and nonparametric classifiers in general) can be divided into two streams: i) global approaches that modify the whole training dataset, e.g., adversarial pruning (arXiv:1706.03922, arXiv:1906.03310, arXiv:2003.06121, etc.), and ii) local approaches that study attack for each data point and find appropriate defense for specific classifiers such as 1-NN (arXiv:1811.00525, arXiv:1906.03972, etc.).

Compared to the current literature, we believe that our approach is more general in two significant ways: i) we start from a generic min-max estimation problem, and our ideas and methodology are easily applicable to other non-parametric settings, and ii) we allow for perturbations on $Y$ to hedge against label contamination. We will include this discussion.

**Reviewer 2:** Thank you very much for your feedback. We would like to emphasize that our paper aims to provide a principled approach to robustify nonparametric estimators, our contributions include the proposal of a novel adversarial estimation framework along with theoretical insights.

Regarding the experiment: the MNIST dataset is still the field's standard benchmark dataset to evaluate and compare performance among models (Google Scholar indicates $\sim$591 citations to the MNIST dataset since 2019 alone). To study how robust a deep learning model is subject to (possibly adversarial) distributional shift, the MNIST dataset is also one of the leading benchmarks (arXiv:1906.02530). State-of-the-art research on robustifying nonparametric estimators (arXiv:1706.03922, arXiv:1906.03310, arXiv:2003.06121, arXiv:1811.00525, arXiv:1906.03972, etc.) also focus on simple experimental settings to condense and deliver insights.

Regarding the performance: From Table 2, our estimator outperforms the N-W estimator from 9% ($N = 500$) to 20% ($N = 50$) in terms of accuracy in the MNIST dataset. Further results in the appendix show that we can stochastically dominate other nonparametric approaches in both synthetic (Figure A.2) and MNIST dataset with $p \leq 1$ (Figure A.3).

**Reviewer 3:** Thank you for your suggestion on the lower bound. Currently we focus on the upper bound because it is the canonical analysis for minimax problems. The probabilistic *lower* bound can be stated in the below result.

**Proposition.** *Under the same settings of Proposition 3.1, with a probability of at least $1 - O(N^{-c})$, we have*

$$\mathbb{E}_{\mathbb{P}}[\ell(Y, \beta^\star)|X \in \mathcal{N}_\gamma(x_0)] \geq - \sup_{\mathbb{Q} \in \mathbb{B}_\rho, \mathbb{Q}(X \in \mathcal{N}_\gamma(x_0)) > 0} \mathbb{E}_{\mathbb{Q}}\big[-\ell(Y, \beta^\star)|X \in \mathcal{N}_\gamma(x_0)\big]$$

To evaluate the supremum on the right-hand side, it suffices to use Theorem 2.3 in the paper with minor changes to the definition of the values $v_i^\star(\beta^\star)$. The proof of this claim follows similar argument as in the proof of Proposition 4.1.

Regarding Example 3.3: Thank you for noticing the linearity condition. We recheck [29, Theorem 2 and Corollary 3] where 'global' consistency is obtained universally in probability. Since we focus on 'local' consistency, only *continuity* of the regression function is required, that means Example 3.3 is valid when $\mathbb{E}_{\mathbb{P}}[Y|X = x]$ is continuous in $x$. We will correct this condition in the revision and clarify the (stronger) notion of consistency that we have in mind.

Regarding the performance of our estimator: In the experiment, we specifically tune our estimator so that it behaves as a robust $k$-nearest neighbor estimator. As such, it is more reasonable to compare our estimator versus the vanilla $k$-nearest neighbor approach. Experiments using both synthetic and the MNIST dataset show that we clearly outperform the $k$-nearest neighbor, which justifies the benefit of being robust.

Even when comparing our estimator versus the N-W estimator, Table 2 shows that our estimator outperforms the N-W estimator at all sample sizes. The improvement can be as big as 20% ($N = 50$) in terms of accuracy in the MNIST dataset. In the appendix, we also present additional results showing that our estimator can stochastically dominates other nonparametric approach in both the synthetic setting (Figure A.2) and the MNIST dataset with $p \leq 1$ (Figure A.3). Our estimator thus delivers performance gains at reasonable computational overhead.

[Meta-Review · NeurIPS 2020]

The reviewers agree that this paper presents a novel framework and approach towards robust non-parametric methods. I will therefore accept. Nevertheless, the authors are encouraged to further improve their empirical study.